# AdaStride: Using Adaptive Strides in Sequential Data for Effective Downsampling

## Abstract

The downsampling layer has been one of the most commonly used deep learning (DL) components in sequential data processing due to its several advantages. First, it improves the generalization performance of networks by acting as an information bottleneck, where it extracts task-relevant features and discards others. Second, it reduces data resolution allowing CNN layers to have larger receptive fields with smaller kernel sizes. Third, the reduced data resolution facilitates the use of Transformer networks in case of high-resolution data. Accordingly, there have been many studies on downsampling methods, but they have a limitation in that they apply the same downsampling ratio across a data instance. Using the same downsampling ratio uniformly for an entire data instance does not reflect the fact that the task-relevant information is not uniformly distributed in real data. In this paper, we introduce AdaStride, a downsampling method that can apply adaptively varying downsampling ratios across a sequential data instance given an overall downsampling ratio. Specifically, AdaStride learns to deploy adaptive strides in a sequential data instance. Therefore, it can preserve more information from task-relevant parts of a data instance by using smaller strides for those parts and larger strides for less relevant parts. To achieve this, we propose a novel training method called vector positioning that rearranges each time step of an input on a one-dimensional line segment without reordering, which is used to build an alignment matrix for the downsampling. In experiments conducted on three different tasks of audio classification, automatic speech recognition, and discrete representation learning, AdaStride outperforms other widely used standard downsampling methods showing its generality and effectiveness. In addition, we analyze how our AdaStride learns the effective adaptive strides to improve its performance in the tasks.

## 1 Introduction

Recently, deep learning (DL) has achieved remarkable performance in various machine learning domains such as image classification (Krizhevsky et al., 2012; He et al., 2016a), machine translation (Bahdanau et al., 2015; Vaswani et al., 2017), audio classification (Yoon et al., 2019; Li et al., 2019), and speech recognition (Chan et al., 2015; Gulati et al., 2020; Kim et al., 2022). This is because many DL architectures such as CNN (Fukushima & Miyake, 1982; LeCun et al., 1989), RNN (Rumelhart et al., 1985; Hochreiter & Schmidhuber, 1997) and Transformer (Vaswani et al., 2017) can be easily employed for various types of input and output.

Especially, downsampling layers have brought many benefits when they are used in combination with other DL layers in many sequential processing tasks. For example, in many classification networks (Li et al., 2019; Ma et al., 2021), the downsampling layer was used with CNN layers where it gradually reduces the data resolution while providing several benefits: (1) it improves the generalization performance of the networks by acting as an information bottleneck that preserves task-relevant information and discards other trivial information (Li & Liu, 2019); (2) it reduces the amount of computation because the reduced resolution allows intermediate CNN layers to have virtually larger receptive fields with smaller kernel sizes. Other than CNN, there have been many studies reporting remarkable results and even the state-of-the-art performance by using downsampling layers and Transformer layers (Dhariwal et al., 2020; Gulati et al., 2020; Kim et al., 2022; Karita et al., 2019; Synnaeve et al., 2019; Collobert et al., 2020) together. In these studies, the downsampling

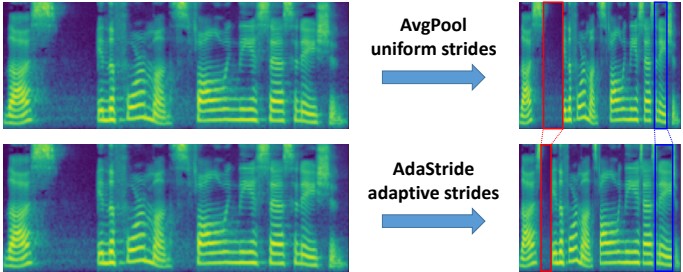

Figure 1: Examples of two different downsampling methods, where they reduce the lengths of melspectrograms in half: (1) average pooling with uniform strides; (2) AdaStride downsampling that uses adaptive strides.

layer was used to alleviate the problem of quadratically increasing memory and computational cost of Transformer, which has hindered Transformer from being used in case of high-resolution data.

Due to the advantages of the downsampling layers, there have been many studies on downsampling (Radford et al., 2016; Zhang, 2019; Rippel et al., 2015; Riad et al., 2022; Zhao & Snoek, 2021; Stergiou et al., 2021) to preserve more task-relevant information or to discard task-interfering information. Radford et al. (2016) replaced the max- or avg- pooling with strided convolutions to improve the performance by allowing the networks to learn their own downsampling. Zhang (2019) pointed out that aliasing can occur in downsampling because the reduced sampling rate decreases the Nyquist frequency (Nyquist, 1928). Therefore, Zhang (2019) proposed to apply low-pass filtering before downsampling to remove the high-frequency components over the Nyquist frequency. Meanwhile, Rippel et al. (2015) proposed spectral pooling that performs downsampling in the frequency domain. Specifically, inspired by the fact that the power spectrum of images mostly consists of lower frequencies, it removes the high-frequency components in the frequency domain. Furthermore, Riad et al. (2022) proposed DiffStride, which improves the performance and narrows down the architecture search space by making the stride factor of the spectral pooling learnable. However, there have been no studies about using varying downsampling ratios across a data instance to fully utilize the given overall downsampling ratio.

In this paper, we introduce AdaStride, a novel downsampling method that applies adaptively varying downsampling ratios in a sequential data sample given an overall downsampling ratio. To be specific, our AdaStride learns to deploy adaptive strides in a sequential data sample. Figure 1 illustrates the usefulness of deploying adaptive strides in a data instance. The upper part of Figure 1 shows average pooling using uniform strides, which inevitably uniformly reduces the data resolution in half for every part of the data. On the contrary, as shown in the lower part of Figure 1, deploying adaptive strides allows for preserving more information for task-relevant parts by using smaller strides for that parts and larger strides for less important parts. To achieve this, we propose a novel training method called vector positioning (VP), which rearranges each time step of the input signal on a one-dimensional line segment without reordering. Based on VP, AdaStride constructs an alignment matrix and performs downsampling based on the matrix. In addition, we also introduce a variant of AdaStride called AdaStride-F, which speeds up the computation by replacing the matrix multiplication in the downsampling with a faster scatter_add operation. This reduces the training time of AdaStride by around 13% while showing similar performance compared to AdaStride.

To evaluate AdaStride, we compare our method with widely used downsampling methods in three different tasks: audio classification, automatic speech recognition (ASR), and discrete representation learning based on VQ-VAE (Van Den Oord et al., 2017). In audio classification, AdaStride outperforms all the standard downsampling methods including DiffStride on three of four datasets without hyperparameter tuning. Also, we identify that it is possible to obtain better performance of AdaStride by adjusting the trade-off between information loss and aliasing depending on the dataset. Indeed, AdaStride ranks first on all of the datasets when we optimize the trade-off independently by tuning a hyperparameter according to the dataset. Furthermore, our AdaStride achieves significant performance improvement over the strided convolution, even when the increase in memory usage of AdaStride is minimized. On other tasks, we show that AdaStride also outperforms all the other downsampling methods by learning effective adaptive strides, and we provide analyses of how the adaptive strides are learned in the tasks.

## 2 BACKGROUND

In this section, we present the background for developing a better understanding of our work. First, we demonstrate how the amount of information that can be preserved in downsampling is bounded, mainly focusing on the relationship between downsampling and aliasing. Then, we will explain spectral pooling based downsampling methods that will appear in comparative experiments, which will also help understand the motivation of our work.

### 2.1 DOWNSAMPLING AND ALIASING

Zhang et al. (2019) showed that commonly used downsampling methods such as max-pooling, average-pooling, or strided convolution can be seen as a combination of '*densely refining an input*' and '*naively resampling the signal*'. For example, let's assume that we perform max-pooling with a window size of $w$ and a stride of $s$. Then, it can be decomposed into the following two steps: (1) applying a max operator of a window size of $w$ to the input signal with a stride of 1; (2) naively resampling the refined input signal with a sampling rate reduced by $1/s$.

In the paper, it was pointed out that aliasing can occur in step (2), and it hurts the shift-invariance and shift-equivariance of networks. Therefore, it proposed to apply low-pass filtering to the signal before step (2), and it actually improved the shift-invariance and shift-equivariance. Furthermore, the low-pass filtering surprisingly even improved the classification performance of networks despite of the information loss for the high-frequency components. It implies that, in downsampling, the components above the Nyquist frequency rather disturb the networks than convey task-relevant information. In other words, it is essential to use a lower downsampling ratio to raise the level at which frequency components become meaningful.

### 2.2 SPECTRAL POOLING AND DIFFSTRIDE

Rippel et al. (2015) proposed spectral pooling that performs downsampling in the frequency domain. Let $\mathbf{x} = \{x_0, ..., x_{T-1}\} \in \mathbb{R}^T$ be the input and $s$ be a stride factor. Then, spectral pooling first sends $\mathbf{x}$ to the frequency domain based on Discrete Fourier Transform (DFT) as $\mathbf{y} = \mathcal{F}(\mathbf{x}) \in \mathbb{C}^{\lfloor \frac{T}{2} \rfloor + 1}$ as follows:

$$y_n = \sum_{t=0}^{T-1} x_t \cdot e^{-i(\frac{2\pi n}{T})t}, \qquad 0 \leq n \leq \left\lfloor \frac{T}{2} \right\rfloor. \tag{1}$$

Then, $\mathbf{y}$ is cropped by maintaining only the lower $\lceil \frac{T}{2s} \rceil + 1$ frequency components, and it is brought back to the original domain through the inverse DFT as $\hat{\mathbf{x}} = \mathcal{F}^{-1}(\text{crop}(\mathbf{y}))$. Inspired by the fact that the power spectrum of natural images mostly consists of low-frequency components, spectral pooling could improve the network performance, and the aliasing problem is naturally resolved here.

Furthermore, Riad et al. (2022) proposed DiffStride that rather applies soft mask $\mathbf{m} \in \mathbb{R}^{\lfloor \frac{T}{2} \rfloor + 1}$ to $\mathbf{y}$ than directly crops $\mathbf{y}$ as follow:

$$m_n = \min\left[\max\left[\frac{1}{R}(R + \frac{T}{2s} + 1 - n),\ 0\right],\ 1\right], \qquad \hat{\mathbf{y}} = \mathbf{m} \odot \mathbf{y}, \tag{2}$$

where $R$ is a hyperparameter and $\odot$ denotes element-wise multiplication. Then, DiffStride crops $\hat{\mathbf{y}}$ by maintaining only the non-zero frequency components and the downsampled signal is obtained as $\hat{\mathbf{x}} = \mathcal{F}^{-1}(\text{crop}(\hat{\mathbf{y}}))$. By doing so, it allowed the model to learn the stride factor $s$ through gradient descent algorithm, resulting in improved classification performance and smaller architecture search space. However, it is also the same for these two spectral pooling methods that the downsampling ratio determines the level at which frequency components become meaningful.

## 3 METHODOLOGY

In this section, we introduce AdaStride, a novel downsampling method that can adaptively apply different downsampling ratios to different parts of a sequential input based on vector positioning process. Also, we introduce a variant of AdaStride called AdaStride-F, which is an efficient version of AdaStride that performs downsampling based on scatter_add operation instead of matrix multiplication.

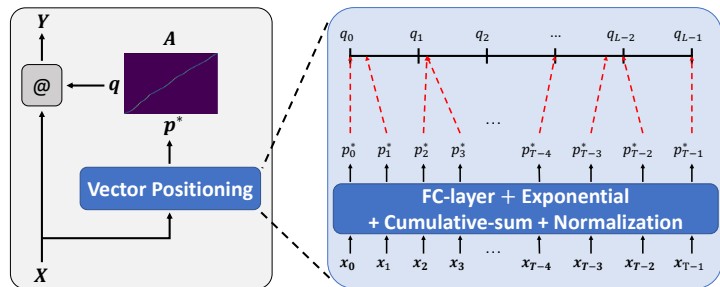

Figure 2: An abstract diagram about how AdaStride downsampling is performed. It is performed based on an alignment matrix, and the matrix is calculated based on the vector positioning process.

## 3.1 ADASTRIDE

Let $\mathbf{X} = \{\mathbf{x}_0, ..., \mathbf{x}_{T-1}\} \in \mathbb{R}^{T \times D}$ be the input of AdaStride and $s$ be a stride factor. Then, AdaStride reduces the time steps of $\mathbf{X}$ by the factor of $s$ and outputs $\mathbf{Y} = \{\mathbf{y}_0, ..., \mathbf{y}_{L-1}\} \in \mathbb{R}^{L \times D}$, $L = \lceil T/s \rceil$ based on a vector positioning (VP) process. Inspired by Efficient-TTS (Miao et al., 2021), we developed the VP where we first define a one-dimensional line segment represented by a list of indices $\mathbf{q} = [0, 1, ..., L-1] \in \mathbb{R}^L$ as shown in Figure 2. Then, unlike Efficient-TTS that aligns the text and speech modalities based on the cross-attention mechanism, each input vector $\mathbf{x}_t$ is positioned on the line segment based only on the input vectors. First, we obtain relevant displacements of the input vectors, $\Delta \mathbf{p} = \{\Delta p_0, ..., \Delta p_{T-1}\} \in \mathbb{R}^T$. Here, each $\Delta p_t$ is calculated by passing each $\mathbf{x}_t$ through a position-wise fully-connected layer followed by an exponential function. Then, the positions of $\mathbf{x}_t$s are calculated using the cumulative summation operation, and they are lastly normalized to be positioned between $[0, \ L-1]$ as follows:

$$p_t = \sum_{n=0}^{t} \Delta p_n, \qquad p_t^* = \frac{p_t - p_0}{p_{T-1} - p_0} * (L-1).$$ (3)

In this way, we can guarantee that the positions of the input vectors start from 0 and end at $L-1$ without reordering.

Then, an alignment matrix $\mathbf{A} \in \mathbb{R}^{L \times T}$ is computed based on the distances between $p_j^*$ and $q_i$ ($\mathbf{D} \in \mathbb{R}^{L \times T}$), and $\mathbf{X}$ is downsampled to $\mathbf{Y}$ based on $\mathbf{A}$ as follows:

$$d_{ij} = -\sigma^{-2}(p_j^* - q_i)^2, \qquad a_{ij} = \frac{\exp(d_{ij})}{\sum_{n=0}^{T-1} \exp(d_{in})}, \qquad \mathbf{Y} = \mathbf{A}@\mathbf{X},$$ (4)

where $\sigma$ is a hyperparameter that controls the sharpness of the alignment and @ denotes matrix multiplication. To sum up, $\mathbf{X}$ is downsampled to $\mathbf{Y}$ based on $\mathbf{A}$ calculated in VP, and we name this downsampling method AdaStride because it can be seen as a sub-sampling with adaptive strides for significantly small $\sigma$ value.

In addition, we empirically observed that it is crucial to prevent every stride from being less than one (i.e., doing upsampling) to stabilize VP learning (Appendix D). In other words, every $p_j^*$ should have at most one $q_i$ where the $p_j^*$ is the nearest position to the $q_i$. Therefore, we add an additional position loss to the original loss of each task to satisfy the condition:

$$\mathcal{L}_{pos} = \frac{1}{T-1} \sum_{t=0}^{T-2} \mathbf{max}((p_{t+1}^* - p_t^*) - 1.0, \ 0.0).$$ (5)

All the calculation steps of AdaStride are summarized in algorithm 1 in Appendix A.

## 3.2 ADASTRIDE-F

To reduce the memory and computational cost of AdaStride, we also propose a variant of AdaStride called AdaStride-F that replaces the matrix multiplication based downsampling with a scatter_add

operation based downsampling. In AdaStride-F, distances are calculated only from each $p_j^*$ to its nearest index $q_i$, where the $q_i$ is equivalent to the index obtained by rounding the $p_j^*$:

$$d_{ij} = \begin{cases} -\sigma^{-2}(p_j^* - q_i)^2, & (q_i = \left[ p_j^* \right]) \\ -\infty, & (q_i \neq \left[ p_j^* \right]) \end{cases}. \tag{6}$$

Then, the alignment matrix $\mathbf{A}$ is calculated as (4), but it is not used for matrix multiplication. Instead, since the alignment matrix $\mathbf{A}$ has only one non-zero element in a column, the matrix multiplication in (2) can be calculated using a weighting vector $\mathbf{w} = \{w_0, ..., w_{T-1}\} \in \mathbb{R}^T$ as follows:

$$w_j = \sum_{i=0}^{L-1} a_{ij}, \qquad \mathbf{y}_i = \sum_{n \in C_i} w_n \cdot \mathbf{x}_n, \qquad C_i = \{\, t \in [0, ..., T-1] \mid [p_t^*] = q_i \,\}, \tag{7}$$

where $C_i$ is a set of indices of $p_t^*$s of which the nearest index is $q_i$, and the weighted sum is implemented using the scatter_add operation. All the calculation steps of AdaStride-F are summarized in algorithm 2 in Appendix A.

## 4 EXPERIMENTS

In this section, we describe the experimental results on the superiority of AdaStride as a downsampling method, where we conduct three different tasks to verify its wide applicability: audio classification, automatic speech recognition (ASR), and discrete representation learning based on VQ-VAE. In all of the tasks, we use audio representaions by converting waveforms to melspectrogram features and the conversion parameters are summarized in Table 7 in Appendix C.

### 4.1 AUDIO CLASSIFICATION

**Experimental setup** We used four audio classification datasets: IEMOCAP dataset (Busso et al., 2008) for speech emotion recognition, Speech commands dataset (Warden, 2018) for speech vocabulary recognition, BirdSong dataset (Stowell et al., 2018) for birdsong detection, and TUT urban dataset (Heittola et al., 2018) for acoustic scene classification. For all of the tasks, we used the same architecture consisting of 6-layer ResBlocks (He et al., 2016b), and downsamplings were performed in the first, third, and fifth ResBlocks where the downsampling ratios were 2. For the downsampling methods other than the strided convolution, we first apply an unstrided convolution and then apply the downsampling methods. At the end of the blocks, hidden representations were globally average-pooled to obtain a single vector, and it was fed to a fully-connected classifier. For more details, please refer to Appendix C.

**Results** As shown in Table 1, our AdaStride and AdaStride-F outperformed the other downsampling methods in three of four datasets without hyperparameter tuning. However, in TUT urban dataset, spectral downsampling methods outperformed AdaStride. We attribute this to the characteristics of the TUT urban dataset, of which audios mostly consist of plain background sound recorded at a certain place: (1) the audios have relatively uniform information density (Figure 3-(c)), so it is difficult for AdaStride to fully utilize the adaptive strides; (2) the noisy nature of the TUT urban dataset is advantageous to the spectral downsampling methods, where the high-frequency noise is likely to incur aliasing in downsampling. Namely, the benefits of learning adaptive strides were likely smaller than the benefits of anti-aliasing. In Appendix E, there is an analysis of how effectively AdaStride utilizes the adaptive strides according to datasets, and in Appendix F, there is a discussion about the relationship between learning adaptive strides and anti-aliasing with additional experiments.

**Computation overhead** To minimize the computation overhead of the AdaStride compared to the strided convolution, we trained a smaller model called AdaStride-S, where the AdaStride layer was only used at the most upper downsampling layer and strided convolution layers were used for the rest downsampling layers. In this case, it still outperformed the strided convolution in all datasets and learned effective alignments at the last AdaStride layer (Figure 13), even though the time and memory overhead was only about 10% (Table 10). Especially, it significantly improved the accuracy on the IEMOCAP dataset, indicating there are tasks that benefit specifically much from using AdaStride. In Appendix G, we provide the computation complexities of different downsampling methods with a comparison of memory and time consumption that are actually used.

Table 1: Audio classification accuracies (% mean ± std) of different downsampling methods measured on four datasets. Each experiment was conducted five times with different random seeds.

| Method | Dataset | | | |
| --- | --- | --- | --- | --- |
| | IEMOCAP | Speech commands | TUT urban | BirdSong |
| Strided Conv. | $46.7 \pm 1.9$ | $95.2 \pm 0.1$ | $94.5 \pm 0.3$ | $82.3 \pm 0.1$ |
| Spectral Pool. | $51.4 \pm 1.8$ | $95.2 \pm 0.2$ | $\mathbf{95.4 \pm 0.4}$ | $83.1 \pm 0.3$ |
| DiffStride | $51.1 \pm 1.3$ | $95.4 \pm 0.1$ | $\mathbf{95.4 \pm 0.3}$ | $82.8 \pm 0.2$ |
| AdaStride | $\mathbf{52.2 \pm 1.0}$ | $\mathbf{95.9 \pm 0.1}$ | $95.1 \pm 0.6$ | $\mathbf{83.3 \pm 0.5}$ |
| AdaStride-F | $51.9 \pm 1.2$ | $\mathbf{95.9 \pm 0.1}$ | $95.1 \pm 0.5$ | $\mathbf{83.3 \pm 0.4}$ |
| AdaStride-S | $51.0 \pm 1.9$ | $95.4 \pm 0.1$ | $94.6 \pm 0.5$ | $82.9 \pm 0.4$ |

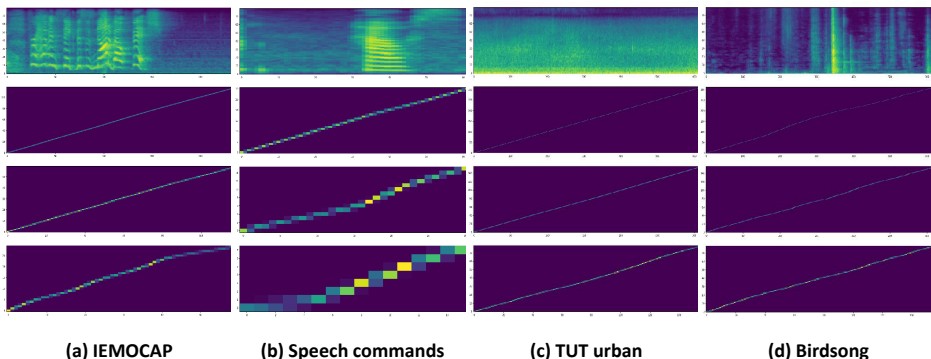

| (a) IEMOCAP | (b) Speech commands | (c) TUT urban | (d) Birdsong |

Figure 3: Each melspectrogram in the first row is a sample from each dataset, and the alignments below a melspectrogram are obtained by AdaStride layers when classifying the melspectrogram.

**Learned alignments**  Figure 3 shows that AdaStride learns effective alignments for classifying audio samples. For example, in Figures 3-(a), (b), AdaStride applies much more downsampling ratios for the silence part and less downsampling ratio for the speech part. By doing so, AdaStride can preserve much more information from the speech part, which contains most of the task-relevant information. In contrast, the alignments in Figure 3-(c) have the shape of an almost straight diagonal line, and it means that AdaStride applies almost the same downsampling ratio to the entire input. Therefore, there is less gain from using AdaStride for the dataset such as the TUT urban dataset.

**Relationship between $\sigma$ and aliasing**  In equations (4) and (6), the hyperparameter $\sigma$ controls the sharpness of the alignment, where the smaller the $\sigma$ is, the sharper the alignment matrix is. In other words, we can balance the trade-off between preserving high-frequency components and anti-aliasing by adjusting the $\sigma$ value. Therefore, we conducted an experiment in which we trained various AdaStride models with different $\sigma$ values on the audio classification tasks. From the experiment, we found out that it was crucial to select an appropriate $\sigma$ value depending on which dataset was used. For example, AdaStride with $\sigma^{-2} = 0.1$ showed the best performance on the TUT urban dataset and even outperformed the spectral downsampling methods, unlike the results in Table 1. Meanwhile, AdaStride with $\sigma^{-2} = 5.0$ achieved the best performance on the IEMOCAP and Birdsong datasets, which means that preserving high-frequency components is preferred to anti-aliasing on those datasets compared to the TUT urban dataset. To see the shape of the alignments learned for different $\sigma$ values, please refer to Figure 14.

## 4.2 AUTOMATIC SPEECH RECOGNITION

**Experimental setup**  In this experiment, we compare various downsampling methods in ASR by using them in a subsampling module to reduce the time resolution of an input. Specifically, the subsampling module consists of two (Conv-Pool-SiLu) blocks where each block downsamples the

Table 2: Audio classification accuracies (% mean $\pm$ std) measured on four datasets using different $\sigma$ values. Each experiment was conducted five times with different random seeds.

| | Dataset | | | |
| --- | --- | --- | --- | --- |
| | IEMOCAP | Speech commands | TUT urban | Birdsong |
| $\sigma^{-2} = 0.1$ | $50.4 \pm 1.7$ | $93.0 \pm 0.1$ | $\mathbf{95.6 \pm 0.3}$ | $76.1 \pm 0.6$ |
| $\sigma^{-2} = 1.0$ | $52.1 \pm 0.7$ | $95.6 \pm 0.1$ | $95.2 \pm 0.5$ | $82.3 \pm 0.7$ |
| $\sigma^{-2} = 5.0$ | $\mathbf{52.2 \pm 1.0}$ | $95.9 \pm 0.1$ | $95.1 \pm 0.6$ | $\mathbf{83.3 \pm 0.5}$ |
| $\sigma^{-2} = 10.0$ | $52.2 \pm 1.0$ | $\mathbf{96.0 \pm 0.1}$ | $95.2 \pm 0.6$ | $83.3 \pm 0.2$ |

input by 2. In the case of an experiment on DiffStride, the experiment was conducted by using a DiffStride block with a spectral pooling layer block to fix the total downsampling ratio to 4. Then, the output of the subsampling module is fed to an encoder consisting of the 16 Squeezeformer (Kim et al., 2022) blocks. We trained ASR models with different downsampling methods based on CTC loss (Graves et al., 2006) on LibriSpeech ASR corpus (Panayotov et al., 2015), and we compared phoneme error rate (PER) and word error rate (WER) of the models. We used g2p module (Park (2019)) for PER models and a 1k subword tokenizer for WER models, which is trained on the train splits using WordPiece algorithm (Wu et al., 2016). Also, we used SpecAugment (Park et al., 2019) with one frequency-mask with mask parameter ($F = 27$) and five time-masks with maximum time-mask ratio ($p_S = 0.05$). For more details such as audio representation and training parameters, please refer to Appendix C.

Table 3: PER and WER (%) of different downsampling methods measured on each data split of the LibriSpeech dataset. A data split with '-other' consists of speakers that are more difficult to recognize.

| | dev-clean | | dev-other | | test-clean | | test-other | |
| --- | --- | --- | --- | --- | --- | --- | --- | --- |
| Method | PER | WER | PER | WER | PER | WER | PER | WER |
| Strided Conv. | 1.46 | 4.87 | 4.66 | 12.29 | 1.55 | 4.88 | 4.43 | 11.91 |
| Spectral Pool. | 1.53 | 4.54 | 4.79 | 11.63 | 1.59 | 4.62 | 4.62 | 11.53 |
| DiffStride | 1.48 | 5.02 | 4.78 | 12.88 | 1.60 | 4.93 | 4.62 | 12.81 |
| AdaStride | 1.40 | **4.05** | **4.56** | **10.98** | **1.47** | **4.11** | 4.28 | **10.75** |
| AdaStride-F | **1.38** | 4.50 | 4.62 | 11.71 | 1.53 | 4.38 | **4.26** | 11.50 |

**Results** As shown in Table 3, our AdaStride and AdaStride-F outperformed the other downsampling methods for both PER and WER for all data splits while combined with the state-of-the-art ASR Squeezeformer architecture. Since recognizing a speech is to extract a word sequence (sequential data) from an audio input (time series data), applying varying downsampling ratios to the audios in ASR is effective rather than hurting the performance. Moreover, the alignments that AdaStride learns show that it learns effective adaptive strides even if SpecAugment is used (Figure 16). In addition, we conducted the ASR experiments in a low-resource setting for more extensive comparisons, where we used only a small fraction of the LibriSpeech dataset as much as 10% and 1%, cf. Appendix H.

## 4.3 DISCRETE REPRESENTATION LEARNING

In this section, we demonstrate an experiment conducted based on VQ-VAE (Van Den Oord et al., 2017) to see whether AdaStride is superior in discrete representation learning over the other downsampling methods. Here, we compare AdaStride with strided convolution and spectral pooling, all of which use a fixed overall downsampling ratio. VQ-VAE is one of the representation learning frameworks that learns to represent data with discrete codes as shown in Figure 4.

**Experimental setup** In this experiment, we used LJSpeech-1.1 dataset (Ito & Johnson, 2017), and we adopted 6-ResBlock Encoder-Decoder architecture as our baseline model, where downsamplings are conducted alternately in the encoder ResBlocks, and nearest neighbor upsamplings are conducted alternately in the decoder ResBlocks. As a result, it learns to obtain code representations of which time steps are reduced by 1/8 from the original time steps of a melspectrogram. When we used AdaStride,

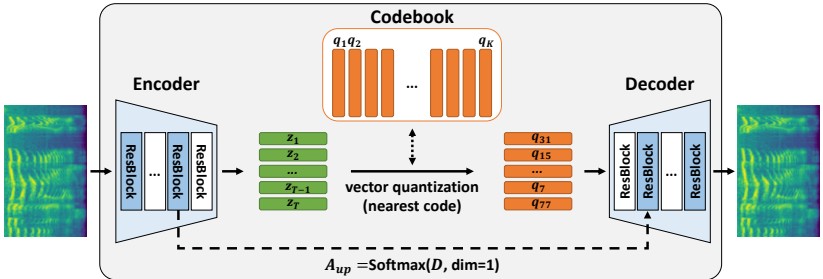

Figure 4: A schematic diagram of the VQ-VAE architecture. (Down- / Up-) samplings are conducted in the colored ResBlocks existing in the encoder and decoder. When AdaStride downsampling layer is used, an alignment matrix is also passed from the encoder to the decoder.

we used it only for the downsampling conducted in the second to last encoder ResBlock for simplicity. This is because AdaStride encodes parts of information using the alignment matrix so the alignment information should be passed to the decoder as well as the discrete code representations. Specifically, we passed an alignment matrix $\mathbf{A_{up}}$ to the second decoder ResBlock for upsampling where $\mathbf{A_{up}}$ was calculated by applying softmax to the distance matrix $\mathbf{D}$ along the first dimension. Therefore, AdaStride-F was excluded in this experiment because applying softmax operation to $\mathbf{D}$ along the first dimension is meaningless for AdaStride-F. For more details such as audio representation, training parameters, architecture, and training of VQ-VAE, please refer to Appendix B and C.

Table 4: PER-R and PER-C (%) for comparing VQ-VAE models with different downsampling methods. PER-R represents a performance of an ASR model trained on reconstructed audios. PER-C represents a performance of an ASR model trained on discrete code representations.

| Method | PER-R | PER-C |
|---|---|---|
| Strided Conv. | 11.44 | 21.23 |
| Spectral Pool. | 18.14 | 25.53 |
| AdaStride | **10.85** | **19.51**[*] / 26.76[**] |

[*], [**]: each of these indicates whether it uses alignments or not

**Results** To see the effectiveness of AdaStride in discrete representation learning, we evaluated the representations that the VQ-VAE models using different downsampling methods learn. First, we evaluated the performance of VQ-VAE itself by comparing the reconstruction performance based on phoneme error rate (PER-R). For this, we generated a hundred audio samples by applying HiFi-GAN (Kong et al., 2020) to the melspectrograms reconstructed by each VQ-VAE model. Then, we measured PER-R on the generated audio samples via the Google Cloud Speech-to-Text API.[1] Second, we evaluated the quality of the discrete representations by training ASR models using the learned discrete representations and measuring phoneme error rate (PER-C). In the case of AdaStride, we separated the cases of whether the alignments that AdaStride learns are used or not used in ASR training. Specifically, the representations are upsampled by two based on the learned alignments or by repeating each representation twice as shown in Figure 8 in Appendix C. The result of the experiment is shown in Table 4, where our AdaStride outperforms the strided convolution and spectral pooling in terms of both PER-R and PER-C. However, since the code representations and alignments are not learned targeting ASR, PER-C becomes worse when the alignments are not used in ASR training. To see the alignments that AdaStride learns, please refer to Figure 15.

**Encoding data with codes and durations** To see how AdaStride is effective compared to the other downsampling methods, we analyzed how the data encoding is learned in terms of codes and their durations. Here, durations of codes mean lengths of consecutive groups of equivalent codes when codes are upsampled back to the length right before the last downsampling. For AdaStride, the code durations were calculated after upsampling the codes using the alignment matrix which was converted

---

[1]https://cloud.google.com/speech-to-text

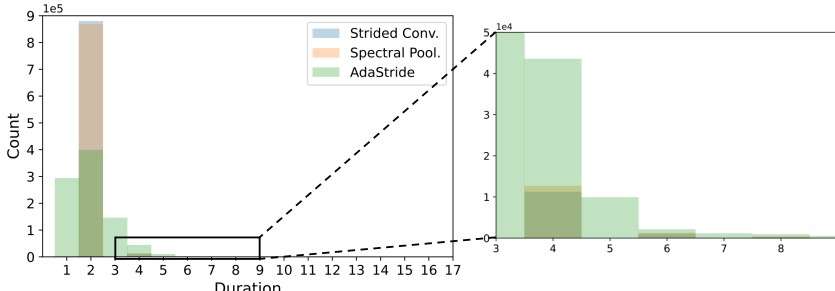

Figure 5: The counts of code durations when VQ-VAE models with different downsampling methods encode the LJSpeech-1.1 dataset. The durations are calculated after the codes are upsampled to the original length, so the durations are even numbers for strided convolution and spectral pooling.

Table 5: The number of unique codes out of a total of 512 codes when VQ-VAE models with different downsampling methods encode the LJSpeech-1.1 dataset.

|  | Strided Conv. | Spectral Pool. | AdaStride |
|---|---|---|---|
| Unique codes | 390 / 512 | 383 / 512 | 365 / 512 |

to a one-hot matrix via argmax operation. Figure 5 illustrates the counts of durations of codes that the different downsampling methods learn. As is expected, AdaStride learned to represent data with codes with variable durations so it can encode information-rich parts of data using more codes in finer resolution. In contrast, strided convolution and spectral pooling ineffectively encoded data with codes most of which have durations of two. Furthermore, the fact that there were no durations longer than two represents that even a long-lasting part in a data sample was encoded using various codes. Therefore, this way of encoding can lead to a waste of codebook capacity, and it can result in a performance drop when the size of the codebook decreases. Indeed, Table 5 illustrates how many unique codes existed when each VQ-VAE model encoded the full LJSpeech-1.1 dataset, where AdaStride used the least number of codes.

**Future work** Although it was shown that Adastride has superiority over other downsampling methods, there are still areas to be studied in the future. First, extending AdaStride to the higher dimensional data. For instance, in image classification, it would be beneficial to preserve more information for the target object instead of the background (Appendix I). Second, addressing the difficulty of finding the optimal $\sigma$ value to control the trade-off between preserving high-frequency components and anti-aliasing. For instance, combining an explicit anti-aliasing process with AdaStride (Appendix F). Lastly, improving its efficiency in terms of memory and time consumption. Even though we introduce AdaStride-F, there is still much room for efficiency improvement in terms of memory and computational cost.

## 5 CONCLUSION

In this paper, we proposed a novel downsampling method called AdaStride, which learns to apply adaptively varying downsampling ratios within a sequential data sample via the vector positioning process. In this way, it can preserve information for task-relevant parts of a data sample better than the fixed overall downsampling rate. In audio classification, AdaStride outperformed other standard downsampling methods on three of four datasets without hyperparameter tuning and on all of the datasets with hyperparameter tuning. In addition, even when we minimized the memory increase of AdaStride compared to strided convolution, it still achieved significant performance improvement, especially on the IEMOCAP dataset. In ASR, AdaStride also outperformed the other downsampling methods in terms of both WER and PER for all data subsets. In VQ-VAE, we saw that AdaStride is superior in reconstructing the input melspectrogram due to its effective utilization of codes and durations, and it indeed leads to better representations for sub-tasks such as ASR.

REPRODUCIBILITY

To help reproduce the experiments, we provided a separate paragraph describing the experimental setup for each sub-task. Also, we elaborated on other details such as data preprocessing, training parameters, and the architectural details in appendices with many figures describing the architectures for accurate re-implementation. Furthermore, we provided a source code of AdaStride as supplementary material, where you can reproduce the audio classification experiment using the AdaStride. Since we took great care to modularize the AdaStride layer while writing the code, it is also possible to easily employ the AdaStride in other tasks.

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

# A   ALGORITHMS

---

**Algorithm 1:** Pseudo-code describing how AdaStride works

---

**Data:**
   $\mathbf{X} \in \mathbb{R}^{T \times D}$: input of AdaStride, a sequential data consisting of $T$ time steps
   $s$: overall stride factor that determines the ratio of the length of the input to the output
   $\sigma$: hyperparameter that controls the sharpness of the alignment matrix
**Result:**
   $\mathbf{Y} \in \mathbb{R}^{L \times D}$: output of AdaStride, a sequential data consisting of $L = \lceil T/s \rceil$ time steps
   $\mathcal{L}_{pos}$: position loss that stabilizes the training of AdaStride

```
/* Caculate normalized positions of the input time steps      */
```
$\Delta\mathbf{p} \leftarrow \texttt{Exponential}(\texttt{PositionalFC}(\mathbf{X}));$
$\mathbf{p} \leftarrow \texttt{Cumsum}(\Delta\mathbf{p}, \text{dim}=0);$
$\mathbf{p}^* \leftarrow (\mathbf{p} - \mathbf{p}[0:1])/(\mathbf{p}[T-1:] - \mathbf{p}[0:1]) * (L-1);$

```
/* Construct an alignment matrix based on p* and q            */
```
$\mathbf{q} \leftarrow \texttt{range}(0, L);$
$\mathbf{D} \leftarrow -\sigma^{-2}(\mathbf{p}^* - \mathbf{q})^2.\text{transpose}();$
$\mathbf{A} \leftarrow \texttt{Softmax}(\mathbf{D}, \text{dim}=1);$

```
/* Perform downsampling and calculate the position loss       */
```
$\mathbf{Y} \leftarrow \mathbf{A}@\mathbf{X};$
$\mathcal{L}_{pos} \leftarrow \frac{1}{T-1}\sum_{i=0}^{T-2} \textbf{max}((\mathbf{p}^*[i+1] - \mathbf{p}^*[i]) - 1.0,\ 0.0)$

---

---

**Algorithm 2:** Pseudo-code describing how AdaStride-F works

---

**Data:**
   $\mathbf{X} \in \mathbb{R}^{T \times D}$: input of AdaStride, a sequential data consisting of $T$ time steps
   $s$: overall stride factor that determines the ratio of the length of the input to the output
   $\sigma$: hyperparameter that controls the sharpness of the alignment matrix
**Result:**
   $\mathbf{Y} \in \mathbb{R}^{L \times D}$: output of AdaStride, a sequential data consisting of $L = \lceil T/s \rceil$ time steps
   $\mathcal{L}_{pos}$: position loss that stabilizes the training of AdaStride

```
/* Caculate normalized positions of the input time steps      */
```
$\Delta\mathbf{p} \leftarrow \texttt{Exponential}(\texttt{PositionalFC}(\mathbf{X}));$
$\mathbf{p} \leftarrow \texttt{Cumsum}(\Delta\mathbf{p}, \text{dim}=0);$
$\mathbf{p}^* \leftarrow (\mathbf{p} - \mathbf{p}[0:1])/(\mathbf{p}[T-1:] - \mathbf{p}[0:1]) * (L-1);$

```
/* Construct an alignment matrix based on p*                   */
```
$\mathbf{d} \leftarrow -\sigma^{-2}(\mathbf{p}^* - [\mathbf{p}^*])^2;$
$\mathbf{D} \leftarrow -\infty(L, T);$
$\mathbf{D} \leftarrow \mathbf{D}.\text{scatter}(\text{dim}=0, \text{index}=[\mathbf{p}^*].\text{t}(), \text{src}=\mathbf{d}.\text{t}());$
$\mathbf{A} \leftarrow \texttt{Softmax}(\mathbf{D}, \text{dim}=1);$
$\mathbf{W} \leftarrow \mathbf{A}.\text{sum}(\text{dim}=0).\text{unsqueeze}(-1);$

```
/* Perform downsampling and calculate the position loss       */
```
$\mathbf{Y} \leftarrow 0(L, D);$
$\mathbf{Y} \leftarrow \mathbf{Y}.\text{scatter\_add}(\text{dim}=0, \text{index}=[\mathbf{p}^*].\text{repeat}(1, D), \text{src}=\mathbf{W} \cdot \mathbf{X});$
$\mathcal{L}_{pos} \leftarrow \frac{1}{T-1}\sum_{i=0}^{T-2} \textbf{max}((\mathbf{p}^*[i+1] - \mathbf{p}^*[i]) - 1.0,\ 0.0)$

---

# B  VQ-VAE

**VQ-VAE**  In this paper, VQ-VAE(Van Den Oord et al., 2017; Razavi et al., 2019) is used to represent a melspectrogram with a sequence of positive integers called codes. During training, the encoder obtains a sequence of continuous vector representations ($z_e(\mathbf{X})$) from a melspectrogram. Then, each of the vector representations is replaced with one among the set of code representations (codebook) based on the nearest neighbor lookup. Lastly, the quantized vector representations ($z_q(\mathbf{X})$) are passed to the decoder, and the decoder learns to reconstruct the original input based on mean absolute error ($\mathcal{L}_{mae}$).

To successfully achieve the discrete representation learning, several techniques are used in VQ-VAE training: (1) during backpropagation, the gradients from the decoder input $z_q(\mathbf{X})$ are passed to the encoder output $z_e(\mathbf{X})$, so encoder can learn to obtain better representations; (2) there are two additional losses used to minimize the gap arisen from the vector quantization process:

$$\mathcal{L}_{codebook}(\mathbf{X}) = \frac{1}{S} \sum_s ||z_q(\mathbf{X})_s - sg[z_e(\mathbf{X})_s]||_2^2, \tag{8}$$

$$\mathcal{L}_{commit}(\mathbf{X}) = \frac{1}{S} \sum_s ||sg[z_q(\mathbf{X})_s] - z_e(\mathbf{X})_s||_2^2, \tag{9}$$

where $S$ is the length of the representations and $sg$ denotes the stop-gradient operation that passes zero gradients during backpropagation.

Furthermore, to prevent the codes from being not used and to speed up training, we adopted several more techniques called code restart and exponential moving average (EMA) code update (Dhariwal et al., 2020). Code restart is a method of replacing a code that is rarely used during training with a representation randomly sampled from $z_e(\mathbf{X})$. EMA code update is to update code representations based on EMA instead of the codebook loss $\mathcal{L}_{codebook}$ to speed up training. As a result, the total loss of VQ-VAE becomes as follows:

$$\mathcal{L}_{vqvae}(\mathbf{X}) = \mathcal{L}_{mae}(\mathbf{X}, G(\mathbf{X})) + \alpha * \mathcal{L}_{commit}(\mathbf{X}), \tag{10}$$

where $G$ is the VQ-VAE model and $G(\mathbf{X})$ is the melspectrogram reconstructed by the model.

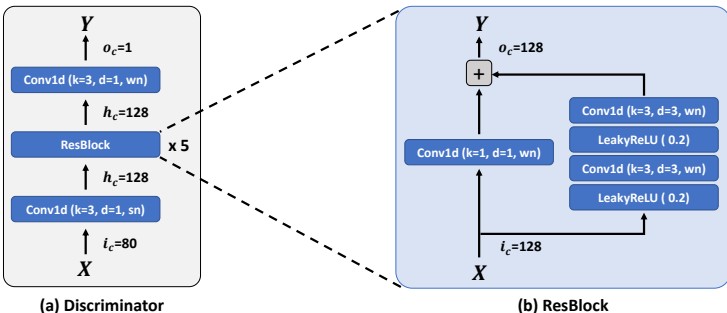

Figure 6: A schematic diagram of the discriminator architecture. **'sn'** denotes spectral normalization (Miyato et al., 2018) and **'wn'** denotes weight normalization (Salimans & Kingma, 2016).

**Adversarial training**  To guide the code representations to capture more perceptually important information, we employed an additional discriminator and adopted the GAN approach in our VQ-VAE training (Esser et al., 2021). We adopted the architecture of the discriminator following (Choi et al., 2021) and it is shown in Figure 6. For GAN loss, we used LSGAN losses (Mao et al., 2017) with additional feature-matching loss, and they are as follows:

$$\mathcal{L}_{adv}(D; G)(\mathbf{X}) = \mathbb{E}_{\mathbf{X}} \left[ (D(\mathbf{X}) - 1)^2 + (D(G(\mathbf{X})))^2 \right], \tag{11}$$

$$\mathcal{L}_{adv}(G; D)(\mathbf{X}) = \mathbb{E}_{\mathbf{X}} \left[ (D(G(\mathbf{X})) - 1)^2 \right], \tag{12}$$

$$\mathcal{L}_{fm}(G; D)(\mathbf{X}) = \mathbb{E}_{\mathbf{X}} \left[ \sum_{l=1}^{5} \frac{1}{N_l} ||D^l(\mathbf{X}) - D^l(G(\mathbf{X}))||_1 \right], \tag{13}$$

$$\tag{14}$$

where $l$ denotes $l$-th layer of the discriminator, and $D_l$ and $N_l$ denote the features and the number of features in the $l$-th layer of the discriminator, respectively. As a result, the final loss of VQ-VAE becomes as follows:

$$\mathcal{L}_D = \mathcal{L}_{adv}(D; G), \tag{15}$$
$$\mathcal{L}_G = \mathcal{L}_{vqvae} + \beta * \mathcal{L}_{adv}(G; D) + \gamma * \mathcal{L}_{fm}(G; D). \tag{16}$$

## C   ADDITIONAL DETAILS

**Dataset**   In Table 6, the information about the datasets used in this paper is demonstrated.

Table 6: Information about the license and about whether the datasets include the personal information or offensive content.

|  | License | Personal information | Offensive content |
|---|---|---|---|
| IEMOCAP | Custom (non-commercial)* | ✗ | ✗ |
| Speech commands | CC-BY | ✗ | ✗ |
| TUT urban | Custom (non-commercial)** | ✗ | ✗ |
| Birdsong | CC-BY 4.0 | ✗ | ✗ |
| LibriSpeech | CC-BY 4.0 | ✗ | ✗ |
| LJSpeech-1.1 | CC0 | ✗ | ✗ |

*: https://sail.usc.edu/iemocap/Data_Release_Form_IEMOCAP.pdf
**: https://zenodo.org/record/1237793/files/LICENSE?download=1

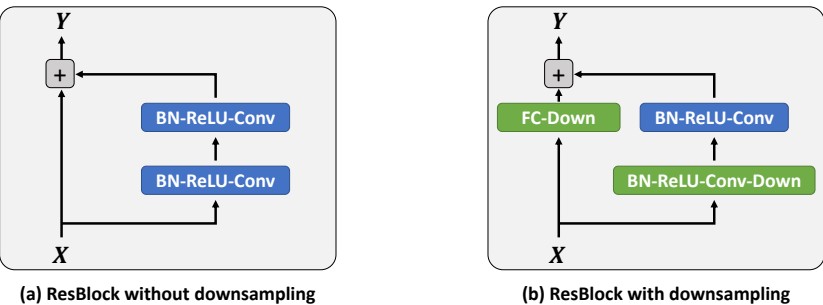

Figure 7: A schematic diagram of the ResBlock architectures with and without downsampling.

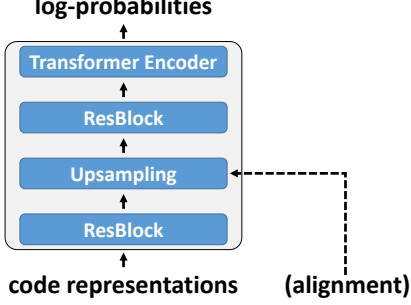

Figure 8: A schematic diagram of the ASR network architecture trained with code representations.

**Architecture of ResBlock**   We adopted PreAct-ResNet block (He et al., 2016b) shown in Figure 7 for audio classification and discrete representation learning. Specifically, we compare various downsampling methods by replacing the strided convolution of the ResNet block with a normal convolution followed by one of the downsampling layers, as shown in Figure 7. When the downsampling

layer is AdaStride, the vector positioning process is conducted in the main path of ResNet, and in the shortcut path of ResNet, downsampling is conducted using the alignment matrix obtained in the main path.

**ASR network architecture trained with code representations**    We trained ASR networks (Figure 8) with code sequences obtained via VQ-VAE encoder based on CTC loss. Also, we optionally used alignments that were obtained by AdaStride Encoder, and if alignments were not used, we expanded the code sequences by two based on nearest neighbor upsampling to match the time resolution of the code sequences equally for a fair experiment. Each network was trained for 100k iterations with a batch size of 32 and a learning rate of 1e-3 using AdamW optimizer (Loshchilov & Hutter, 2019).

**Training instability and layer normalization**    In ASR and VQ-VAE training, we sometimes found out that it was unstable to train models using AdaStride due to the exponential function existing in the vector positioning process. However, we could easily resolve this problem by putting a layer normalization layer (Ba et al., 2016) before the AdaStride layer.

**Hyperparameters**    In Table 7, the hyperparameters used in each task are demonstrated.

Table 7: Hyperparameters used in each task.

|  | Audio classification | Speech recognition | VQ-VAE |
|---|---|---|---|
| hidden dimension | 256 | 144 | 512 |
| kernel size | 3 | 3 | 3 |
| $\sigma^{-2}$ | 5.0 | 5.0 | 10.0 |
| K (# of codes) | - | - | 512 |
| $\alpha$ | - | - | 0.1 |
| $\beta$ | - | - | 1.0 |
| $\gamma$ | - | - | 1.0 |
| sampling rate | 16000 | 16000 | 22050 |
| filter banks | 80 | 80 | 80 |
| windows | 1024 | 400 | 1024 |
| hops | 256 | 160 | 256 |
| optimizer | Adam[*] | AdamW[**] | Adam |
| learning rate | 1e-4 | 5e-4 | 3e-4 |
| batch size | 128 | 256 | 64 |

[*]: Kingma & Ba (2015)
[**]: Loshchilov & Hutter (2019)

# D ABALATION STUDY ON THE POSITION LOSS

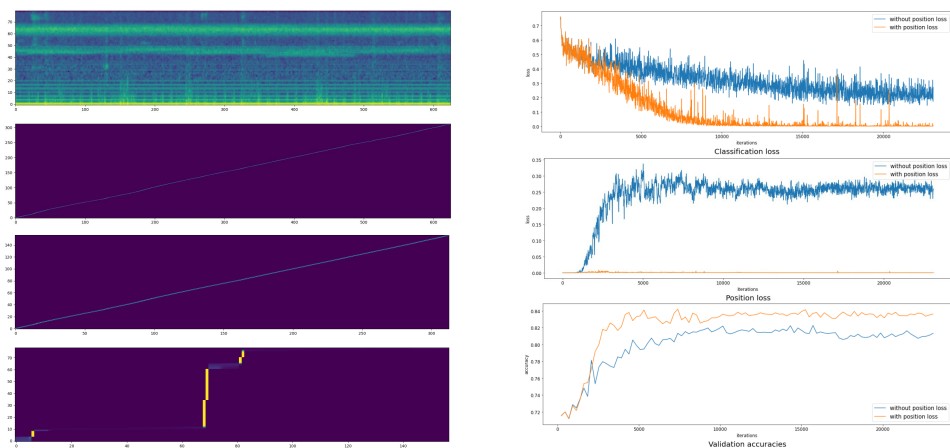

Figure 9: Learned alignments and the training losses and validation accuracy curves during training of the Birdsong classification task when position loss is used or not.

We empirically observed that, witout position loss, training of the models becomes unstable leading to radical alignment learning and poor performance. For instance, Figure 9 shows the training logs when a ResNet equipped with AdaStride layers is trained on the Birdsong dataset: (1) the alignments that AdaStride layers learn; (2) training losses and validation accuracy curves when position loss is used or not.

# E EFFECTIVENESS OF ADAPTIVE STRIDES ACCORDING TO DATASETS

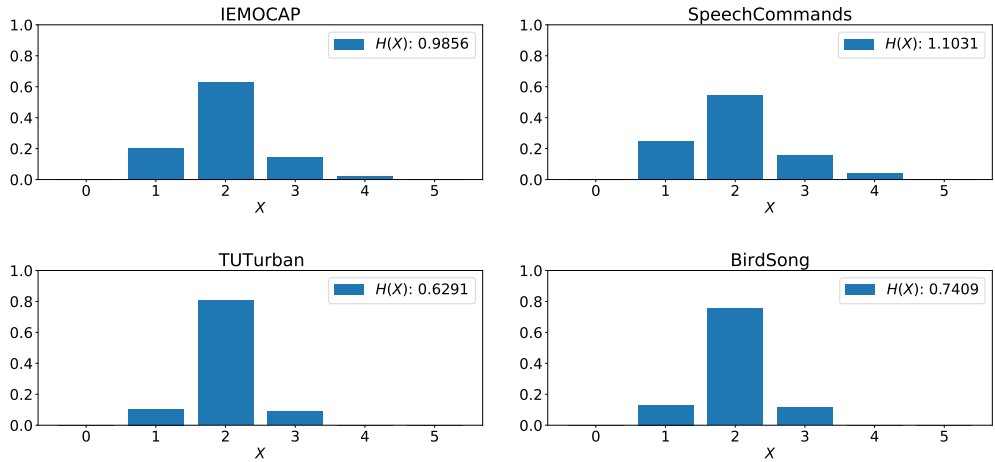

Figure 10: Approximated probability mass functions of the number of frames that are pooled into a single frame. These are approximated by counting the numbers of the pooled frames after applying argmax operation to the learned alignments. On the upper right corner, an entropy calculated from each approximated probability mass function is written.

To see how effectively AdaStride utilizes the adaptive strides according to different datasets, we counted the numbers of frames that are pooled into a single frame after applying the argmax operation to the learned alignments. Herein, the alignments are obtained from the test splits of each audio classification dataset after training the models using the AdaStride layer for the uppermost downsampling

layer. Figure 10 shows the probability mass functions of the number of the pooled frames which are approximated based on the counts, with the entropies written on the upper right corner. As we guessed in Section 4.1, the model trained on the TUT urban dataset utilized the adaptive strides to the smallest extent, indicating that it does not utilize the adaptive strides effectively.

# F   ADAPTIVE DOWNSAMPLING AND ANTI-ALIASING

Table 8: Audio classification accuracies (% mean $\pm$ std over 5 runs) of LPFAdaStride measured on four datasets. Each experiment was conducted five times with different random seeds.

|  | Dataset | | | |
| Method | IEMOCAP | Speech commands | TUT urban | BirdSong |
| --- | --- | --- | --- | --- |
| LPFAdaStride | $51.7 \pm 0.8$ | $96.0 \pm 0.1$ | $95.2 \pm 0.6$ | $83.3 \pm 0.4$ |

As mentioned in Section 4.1, both preserving more task-relevant information (adaptive downsampling) and discarding task-interfering information (anti-aliasing) are significantly important for effective downsampling. Therefore, we again conducted the audio classification experiments in Section 4.1 with a model called LPFAdaStride, where we combine the two beneficial design elements. Specifically, we located a low-pass filter that removes the upper-half frequency components from the input representations before an AdaStride downsampling layer. The result of this experiment is shown in Table 8. Although we expected that the accuracies would significantly increase based on the adaptive downsampling and anti-aliasing that work complementarily, they did not. We attribute this to the varying Nyquist Frequencies according to the adaptive strides so the frequency threshold of the low-pass filters should also be set adaptively according to the adaptive strides.

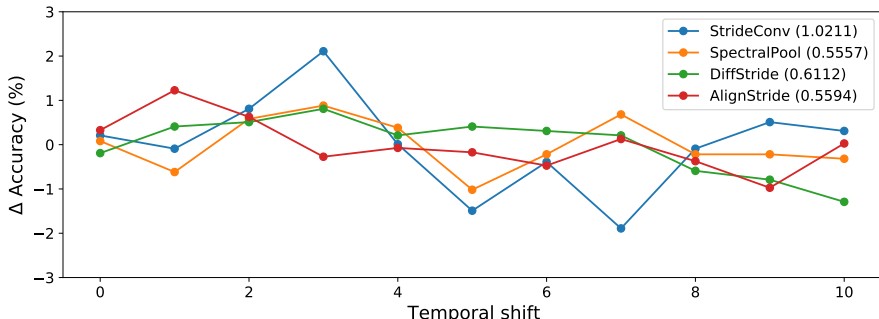

Figure 11: A graph of the accuracy changes based on the average accuracy when shifting the input melspectrogram from 0 to 10 time frames. On the upper right corner, the standard deviations of the accuracy changes are written for different downsampling methods.

In addition, we compared how the accuracies of different downsampling methods change on IEMO-CAP dataset when shifting the input melspectrograms. We conducted this experiment to see how invariant AdaStride is to the input shift as was done in Zhang et al. (2019). The result is shown in Figure 11. In this experiment, AdaStride ranked second in terms of robustness based on the standard deviation of the accuracy changes, which indicates that the distance-based soft downsampling of AdaStride indeed has some anti-aliasing effect.

## G  MEMORY AND TIME CONSUMPTION

Table 9: Computation complexities of the downsampling methods. $n$ is the sequence length and $d$ is the hidden dimension.

|  | Strided Conv. | Spectral Pool. | DiffStride | AdaStride | AdaStride-F |
|---|---|---|---|---|---|
| Complexity | $O(n \cdot d)$ | $O(n \log n \cdot d)$ | $O(n \log n \cdot d)$ | $O(n^2 \cdot d)$ | $O(n \cdot d)$ |

Table 9 shows the computational complexities of the downsampling methods. Strided convolution has the computation complexity of $O(n \cdot d)$. Spectral pooling and DiffStride have the same computation complexity of $O(n \log n \cdot d)$, which corresponds to the Fourier transform. AdaStride has the computation complexity of $O(n^2 \cdot d)$, which corresponds to the matrix multiplication between the alignment matrix and the input representations. AdaStride-F has the computation complexity of $O(n \cdot d)$, which corresponds to the position-wise linear layer and the scatter_add operation.

Table 10: Memory and time consumption per epoch for different downsampling methods relative to strided convolution.

|  |  | Strided Conv. | Spectral Pool. | AdaStride | AdaStride-F | AdaStride-S |
|---|---|---|---|---|---|---|
| Training | Mem | 1.00 | 1.38 | 2.53 | 2.34 | 1.11 |
|  | Time | 1.00 | 1.25 | 1.38 | 1.20 | 1.08 |
| Inference | Mem | 1.00 | 1.00 | 1.24 | 1.25 | 1.08 |
|  | Time | 1.00 | 1.70 | 1.66 | 1.65 | 1.08 |

In addition, we actually measured the memory and time consumption of the different downsampling methods on the IEMOCAP dataset. Specifically, Table 10 shows the memory and time consumption used to process an epoch for training and inference for different downsampling methods relative to strided convolution.

## H    AUTOMATIC SPEECH RECOGNITION ON LOW-RESOURCE DATASETS

Table 11: PER and WER results (%) of different downsampling methods measured on each data split of the LibriSpeech dataset. A data split with '-other' consists of speakers that are more difficult to recognize. Here, only 10% of the LibriSpeech train dataset is used.

| Method | dev-clean | | dev-other | | test-clean | | test-other | |
|---|---|---|---|---|---|---|---|---|
| | PER | WER | PER | WER | PER | WER | PER | WER |
| Strided Conv. | 5.23 | 20.94 | 12.79 | 35.81 | 5.37 | 21.38 | 12.87 | 36.31 |
| Spectral Pool. | 4.94 | 25.18 | 12.52 | 40.68 | 5.02 | 25.58 | 12.30 | 41.31 |
| DiffStride | 4.73 | 24.75 | 12.06 | 40.19 | 4.78 | 25.07 | 12.03 | 41.20 |
| AdaStride | **4.50** | 15.47 | 11.65 | **29.27** | **4.66** | **15.51** | **11.70** | **29.97** |
| AdaStride-F | 4.56 | **15.40** | **11.64** | 29.29 | 4.78 | 15.61 | 11.76 | 30.16 |

Table 12: PER and WER results (%) of different downsampling methods measured on each data split of the LibriSpeech dataset. A data split with '-other' consists of speakers that are more difficult to recognize. Here, only 1% of the LibriSpeech train dataset is used.

| Method | dev-clean | | dev-other | | test-clean | | test-other | |
|---|---|---|---|---|---|---|---|---|
| | PER | WER | PER | WER | PER | WER | PER | WER |
| Strided Conv. | 28.52 | 69.86 | 39.32 | 77.82 | 28.33 | 70.21 | 39.98 | 78.17 |
| Spectral Pool. | 32.50 | 70.12 | 42.83 | 78.13 | 32.69 | 70.13 | 43.40 | 78.33 |
| DiffStride | 34.06 | 69.51 | 43.67 | 77.72 | 33.78 | 69.75 | 44.51 | 77.79 |
| AdaStride | 21.68 | 65.35 | 33.68 | **74.45** | 21.82 | **65.49** | 34.41 | **75.10** |
| AdaStride-F | **21.53** | **65.33** | **33.31** | 74.53 | **21.46** | 65.96 | **34.25** | 75.12 |

For more extensive experiments, we also did the speech recognition experiments under the low-resource setting, where we used only a small fraction of the LibriSpeech dataset as much as 10% and 1%. In the experiments, AdaStride consistently outperformed the other downsampling methods.

## I    ADASTRIDE FOR IMAGE DATA

Table 13: Image classification accuracies (% mean ± std over 5 runs) of different downsampling methods measured on two datasets. Each experiment was conducted five times with different random seeds.

| Method | Dataset | |
|---|---|---|
| | CIFAR-10 | CIFAR-100 |
| Strided Conv. | 91.6 ± 0.1 | 68.4 ± 0.4 |
| Spectral Pool. | 91.9 ± 0.1 | 68.8 ± 0.2 |
| DiffStride | **92.3 ± 0.1** | **69.5 ± 0.1** |
| AdaStride | 92.1 ± 0.1 | 68.8 ± 0.3 |

Although the concept of adaptively downsampling the input representations can be extended to various domains such as images, it is not straightforward to apply AdaStride in the two-dimensional space. Especially, it is challenging to locate the spatial features onto the two-dimensional plane defined by indices based on the cumulative sum operation while preserving their spatial ordering. This is because, the summation of the relevant displacements along an arbitrary course is not conservative in the two-dimensional space. Nevertheless, we compare the different downsampling methods on the image classification using CIFAR-10 and CIFAR-100 datasets (Krizhevsky et al., 2009).

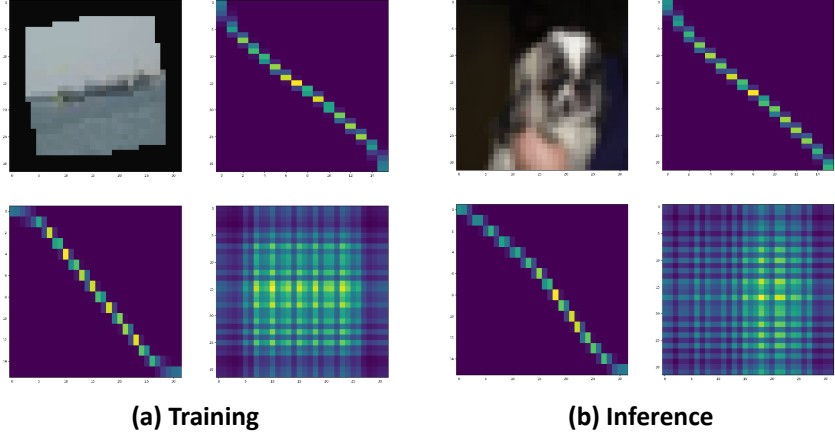

**(a) Training**          **(b) Inference**

Figure 12: Input images and alignments obtained from an AdaStride layer of an image classification model during training and inference, where downsamplings are applied twice sequentially for height and width axes. The alignment matrix shown on the upper right corner is used to downsample the image along the height axis, and the alignment matrix shown on the below left corner is used to downsample the image along the width axis. The image shown on the below right corner is the outer product of the two alignment matrices.

When using AdaStride, we applied it twice sequentially along the height and width axes. Table 10 shows the accuracies of different downsampling methods measured on CIFAR-10, 100 datasets. In the experiment, AdaStride outperformed the strided convolution and spectral pooling but did not outperform the DiffStride. We conjecture that this is because the downsamplings along the different axes happen independently so the effect of learning adaptive downsampling is limited. Actually, in Figure 12, it is shown that the independently learned alignments are not flexible enough to preserve much information from the task-relevant object.

## J  ALIGNMENT SAMPLES

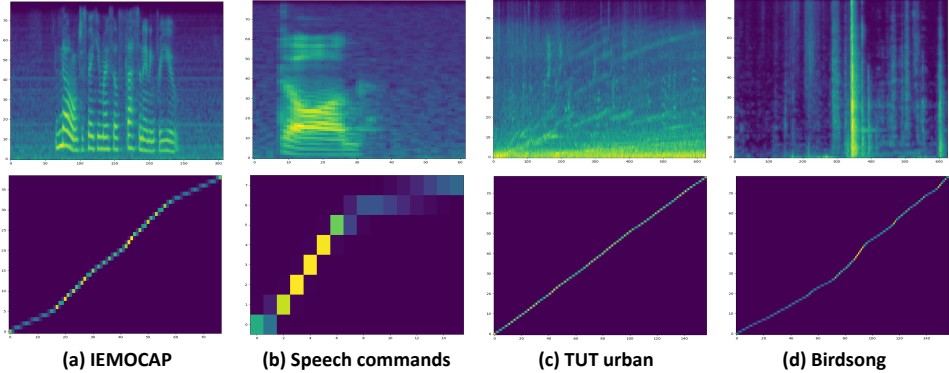

**(a) IEMOCAP**  **(b) Speech commands**  **(c) TUT urban**  **(d) Birdsong**

Figure 13: Each melspectrogram in the first row is a sample from each dataset. Also, the alignment below a melspectrogram is the alignment obtained from an AdaStride layer of an audio classification model when the AdaStride layer is used only for the last downsampling layer.

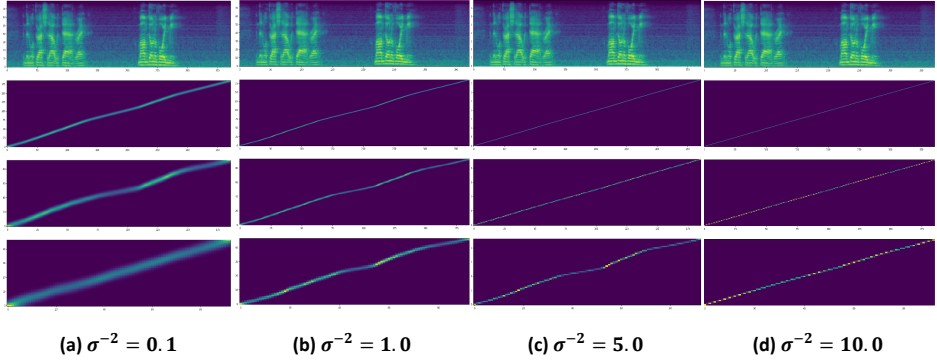

(a) $\sigma^{-2} = 0.1$    (b) $\sigma^{-2} = 1.0$    (c) $\sigma^{-2} = 5.0$    (d) $\sigma^{-2} = 10.0$

Figure 14: Each melspectrogram in the first row is a sample from the IEMOCAP dataset. Also, the alignments below the melspectrogram are the alignments learned by AdaStride layers when classifying the melspectrogram during training. Each column represents different sigma values.

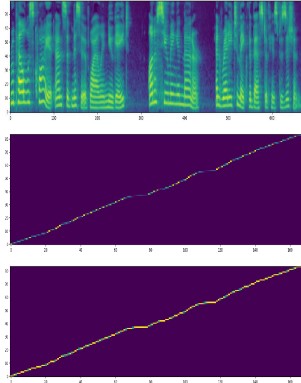

Figure 15: Melspectrogram and alignments obtained from the ResBlocks of the VQ-VAE. The alignment at the bottom is the alignment used for upsampling and it is transposed here.

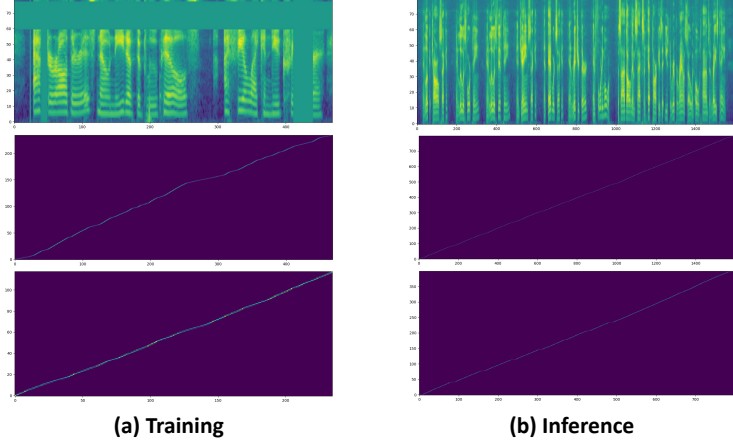

(a) Training    (b) Inference

Figure 16: Melspectrograms and alignments obtained from the AdaStride layers of the ASR networks. During training, SpecAugment is applied to the input features.

