# OpenReview forum: "AdaStride: Using Adaptive Strides in Sequential Data for Effective Downsampling"
_ICLR.cc/2023/Conference — Submitted to ICLR 2023_

### Official Review · Reviewer_PyHD · 2022-10-25

**Confidence:** 2
**Correctness:** 3
**Technical Novelty And Significance:** 3
**Empirical Novelty And Significance:** 3
**Recommendation:** 6

**Clarity, Quality, Novelty And Reproducibility:**

The writing of this paper is generally clear. Personally, I think this paper is of high quality.

However, as a matter of fact, I'm not quite familiar with the area of audio processing. Therefore, I may not be able to properly evaluate the novelty of this paper.

**Strength And Weaknesses:**

Strength:

- The motivation is clear and natural. The proposed method is novel on top of DiffStride.
- Both the empirical results of supervised learning and self-supervised learning are provided.
- AdaStride is a plug-in module, and can be applied to many deep networks.

Weaknesses:

- Only the audio data is considered. Can AdaStride be applied to 2D image data?
- It would be better to present an analysis of the computation cost of AdaStride.


##Post-rebuttal##
After reading the rebuttal from the authors, I decided to keep my score. I think that this paper is just around the borderline and leaning toward accepting. However, as I have mentioned, I'm not quite familiar with the area of audio processing. Therefore, I may not be able to properly evaluate the novelty of this paper.


**Summary Of The Paper:**

This paper an adaptive down-sampling method, AdaStride. The aim of AdaStride is to learn to deploy adaptive strides in a sequential data instance, i.e., preserving more information from task-relevant parts by using smaller strides while using larger strides for less-relevant parts. This idea is implemented by the cumulative-sum + normalization algorithm followed by a soft feature aggregation. Experimental results of speech classification and self-supervised learning are provided.

**Summary Of The Review:**

I think that the quality of this paper is well. My only concern is that I'm not quite familiar with the area of audio processing. Therefore, I may not be able to properly evaluate the novelty of this paper.

---

> ### Author Response · Authors · 2022-11-15
> **Response to PyHD**
>
> Thank you for thinking about our paper as novel and useful and also for giving valuable feedback. We are so glad to be able to deliver our motivation clearly. We have incorporated your feedback into our paper and provided responses to your questions below.
> &nbsp;
> &nbsp;
> **Q: Only the audio data is considered. Can AdaStride be applied to 2D image data?**
> A: Although the concept of adaptively downsampling the input representations can be extended to various domains such as images, it is not straightforward to apply AdaStride in the two-dimensional space. Especially, it is challenging to locate the spatial features onto the two-dimensional plane defined by indices based on the cumulative sum operation, while preserving their spatial ordering. However, in order to answer your question, we added the discussion about applying AdaStride to images with additional experiments in Appendix I, so please refer to it.
> &nbsp;
> &nbsp;
> **Q: It would be better to present an analysis of the computation cost of AdaStride.**
> A: Following your suggestion, we added the theoretical computational complexities in Appendix B, in addition to the actual time and memory cost of the different downsampling methods. Thank you for your valuable feedback and please refer to the update.
> &nbsp;
> &nbsp;
> If your concerns are sufficiently addressed and the updated paper seems to have improved by reflecting the feedback, we appreciate it if you would consider giving us a more positive score.  Otherwise, please let us know about your remaining concerns.

---

> ### Author Response · Authors · 2022-12-10
> **Reponse to your opinion after the rebuttal.**
>
> **Dear PyHD,**
>
> Thank you for reading our rebuttal, considering the quality of our paper well, and also giving your opinion about it.
> Also, we lastly write this response to alleviate any concerns that 'maybe' remain regarding your unfamiliarity with audio processing.
> &nbsp;
> ㅡㅡㅡㅡㅡㅡㅡㅡㅡㅡㅡㅡㅡㅡㅡㅡㅡㅡㅡㅡㅡㅡㅡㅡㅡㅡㅡㅡㅡㅡㅡㅡㅡㅡㅡㅡㅡㅡㅡㅡㅡㅡㅡㅡㅡㅡㅡㅡㅡㅡㅡㅡㅡㅡㅡㅡㅡㅡㅡㅡㅡㅡㅡㅡㅡㅡㅡㅡ
> In order to understand the novelty of AdaStride, you don't have to be very familiar with audio processing because the AdaStride is developed for general sequential data, and the audio data was only used as a representative of sequential data. Therefore, we hope you move away from the perspective of audio data and evaluate the novelty of our AdaStride as a general downsampling method that can be used for any sequential data. Actually, in the experiments, the various audio datasets such as speech and daily recording had different characteristics even within the category of audio data. Therefore, in Appendix E, we conducted the analysis of the effectiveness of AdaStride according to the different datasets. Furthermore, in Appendix I, we also showed that our AdaStride is effective in downsampling images where the attention weights were high near the target object region.
> ㅡㅡㅡㅡㅡㅡㅡㅡㅡㅡㅡㅡㅡㅡㅡㅡㅡㅡㅡㅡㅡㅡㅡㅡㅡㅡㅡㅡㅡㅡㅡㅡㅡㅡㅡㅡㅡㅡㅡㅡㅡㅡㅡㅡㅡㅡㅡㅡㅡㅡㅡㅡㅡㅡㅡㅡㅡㅡㅡㅡㅡㅡㅡㅡㅡㅡㅡㅡ
> &nbsp;
> Thank you again for taking your precious time to review our paper.

---

### Official Review · Reviewer_PVam · 2022-10-26

**Confidence:** 3
**Correctness:** 3
**Technical Novelty And Significance:** 2
**Empirical Novelty And Significance:** 3
**Recommendation:** 5

**Clarity, Quality, Novelty And Reproducibility:**

Some main concerns:

1. The method belongs to pooling methods used for networks. Some related works are missed.
      LiftPool: Bidirectional ConvNet Pooling. J. Zhao and C. Snoek. ICLR 2021.
     Refining activation downsampling with SoftPool. A. Stergiou et. al. ICCV 2021.

2. Except for results gain, it lacks some analysis for shift-invariance and shift-equivariance.

3. It claims that the inferior results on the dataset 'TUT urban' is due to the property of the dataset itself. Then how to show the method's generality. Since it uses adaptive strides, it should also work well for the dataset with uniform information density, right?

4. To illustrate its generality, how the method performs on other kinds of datasets, like image?

4. Besides the results comparisons, computation should also be compared among different methods.

5. The captions for figures and tables lack details.

**Strength And Weaknesses:**

Strength: The paper written is good; the idea is easy to follow; the motivation is clear.

weaknesses: See the next Section.

**Summary Of The Paper:**

This paper proposes a new downsampling method using adaptive strides for sequential data, especially audio data. Specifically, it rearranges each time step of an input on a one-dimensional line segment by using a method called vector positioning. By doing so, it builds an alignment matrix for the downsampling. The paper shows the results of the method on different tasks including audio classification, automatic speech recognition, and discrete representation learning, and claims the method is generalizable and effective.

**Summary Of The Review:**

Please refer to the weaknesses and details above.

---

> ### Author Response · Authors · 2022-11-15
> **Response to PVam**
>
> We are glad to be able to deliver our motivation clearly and thank you for giving valuable feedback on our paper. We have incorporated your feedback into our paper and provided responses to your questions below.
> &nbsp;
> &nbsp;
> **Q: The method belongs to pooling methods used for networks. Some related works are missing. LiftPool: Bidirectional ConvNet Pooling. J. Zhao and C. Snoek. ICLR 2021. Refining activation downsampling with SoftPool. A. Stergiou et. al. ICCV 2021.**
> A: Thank you for your suggestion. We added the citation for the papers that you recommended in the updated paper.
> &nbsp;
> &nbsp;
> **Q: Except for results gain, it lacks some analysis for shift-invariance and shift-equivariance.**
> A: Thank you for your valuable experimental recommendation that can make our paper more solid. In Section F, we added a discussion about learning adaptive strides and anti-aliasing with two additional experiments, where the second experiment is to see the accuracy changes when shifting the input melspectrogram to see how invariant our AdaStride is to the input shift. So, please refer to it.
> &nbsp;
> &nbsp;
> **Q: It claims that the inferior results on the dataset 'TUT urban' are due to the property of the dataset itself. Then how to show the method's generality. Since it uses adaptive strides, it should also work well for the dataset with uniform information density, right?**
> A: The fact that Adastride can not outperform the spectral-based downsampling methods on the TUTurban dataset does not indicate that AdaStride does not benefit from learning adaptive strides because it still outperforms the strided convolution that uses fixed strides. It is just that the benefit of learning adaptive strides was less than anti-aliasing, so we showed it is important to use an appropriate $\sigma$ value for the anti-aliasing in the ablation study. In the updated paper, we added more explanations about this to make it clearer.
> &nbsp;
> &nbsp;
> **Q: To illustrate its generality, how does the method perform on other kinds of datasets, like images?**
> A: Although the concept of adaptively downsampling the input representations can be extended to various domains such as images, it is not straightforward to apply AdaStride in the two-dimensional space. Especially, it is challenging to locate the spatial features onto the two-dimensional plane defined by indices based on the cumulative sum operation, while preserving their spatial ordering. However, in order to answer your question, we added the discussion about applying AdaStride to images with additional experiments in Appendix I, so please refer to it.
> &nbsp;
> &nbsp;
> **Q: Besides the comparisons of the results, computation should also be compared among different methods.**
> A: Following your suggestion, we added the computational complexities in Appendix G, in addition to the actual time and memory cost of the different downsampling methods. Thank you for your valuable feedback and please refer to the update.
> &nbsp;
> &nbsp;
> **Q: The captions for figures and tables lack details.**
> A: Thank you for your suggestion. Reflecting your feedback, we updated the captions of the figures and tables in the paper in more detail.
> &nbsp;
> &nbsp;
> If your concerns are sufficiently addressed and the updated paper seems to have improved by reflecting your feedback, we appreciate it if you would consider accepting our paper.  Otherwise, please let us know about your remaining concerns.

---

> ### Author Response · Authors · 2022-12-13
> **Looking forward to your opinion after our rebuttal!**
>
> Dear reviewer PVam,
>
> Thank you again for your reviews on our paper. In our response above, we have responded to your comments by providing the following results and clarifications.
>
> * Taking your advice, we have included the additional citations that you recommended.
> * We have added a new analysis on the accuracy changes when shifting the input melspectrogram from 0 to 10 time frames to see its robustness to the input shift.
> * We have clarified your concern about why AdaStride does not benefit from learning adaptive strides when the dataset has uniform information density: it wasn't saying that our AdaStride does not benefit from learning adaptive strides, but it was saying that it is not larger than the benefit of anti-aliasing and it still outperformed the strided convolution that uses fixed strides. We have clarified this in the revised paper.
> * We have added new analyses on applying our AdaStride on the image dataset for more discussion in Appendix I to illustrate its generality.
> * We have added theoretical and empirical analyses on computations of different downsampling methods in Appendix G, where we showed that its scalability on long sequences is linear like the strided convolution.
> * We rewrite the captions for figures and tables in more detail.
>
> We are wondering whether these results and clarifications have addressed your questions and concerns. Please express your opinion through comments or by updating your score before the deadline.
>
> Thank you!
>
> Authors

---

### Official Review · Reviewer_MsFg · 2022-10-28

**Confidence:** 3
**Correctness:** 3
**Technical Novelty And Significance:** 2
**Empirical Novelty And Significance:** 3
**Recommendation:** 6

**Clarity, Quality, Novelty And Reproducibility:**

Very clear paper.
Novelty wrt Efficient-TTS is questionable.
Reproducibility is favored by extensive details on hyperparameters and experimental settings in appendix.

**Strength And Weaknesses:**

Strengths:
 * Learning adaptive downsampling is a fundamental deep learning problem with very large potential impact across all types of tasks, models and modalities.
 * This approach displays two characteristics that are offered by strided convs, spectral pooling or DiffStride: 1) the striding is input dependent 2) the striding is uneven along the temporal axis and can emphasize certain segments
 * The algorithm is simple and despite the computational overhead it could be useful in practice
 * It's really nice to see this approach tested in a generative VQ-VAE setting, while the benchmarks are typically classification networks.
 * Lots of details on the experimental setup in appendix

Weaknesses:
 * Model:
  * I suspect that authors do not show how close their approach is to the learnable alignment layer of [Efficient-TTS](http://proceedings.mlr.press/v139/miao21a/miao21a.pdf) which seems to accomplish the same thing (learn an alignment between x and its downsampled version). If the models are indeed very close and authors simply apply it in the context of downsampling, the contribution is still worth publication but they should be honest about that while currently the reference to Efficient-TTS is kind of hidden in the text.
  * Authors should specify more clearly how they replace strided convs by adastride, even though I assume it first applies an unstrided convolution.
  * AdaStride-F is presented as an implementation trick, yet performance between AdaStride and AdaStride-F vary a lot (e.g. for ASR). Why?
  * This approach is conceptually very different from spectral pooling or DiffStride that exploit global spatial information for downsampling, while AdaStride only repeats or removes frames, as would be done in DTW. It would be interesting to have more discussion on this fundamental difference.
 * Experiments:
  * The gains on the audio classification tasks are not significant, and I find the justification for the particularly weak performance on TUT not convincing, or at least incomplete. The authors explain that the "information density is uniform" in acoustic scenes which is somehow incorrect (and speculative anyway) since a car honking or a bird chirping are transient events that can hint at the type of acoustic scene.
  * it is explained that AdaStride-S improves IEMOCAP but this is not what is shown by Table 1
  * The audio classification dataset being quite small and subject to wide variations in results based on confounding factors such as the random seed, I find it not very convincing to cross-validate configurations (from AdaStride to AdaStride-S) or hyperparameters (sigma) on them, in particular on the test set.
  * DiffStride is excluded from ASR experiments due to learning the downsampling factor, why is it a limitation?

Questions:
* Why is Adastride limited to sequential data? What are the technical challenges to extend it to an arbitrary number of dimensions?
* Section 3.1: why is the Miao reference put after the definition of a simple vector?


Typos:
* 4.1: "dowmsampling" -> "downsampling"
* Section 3.1: "a exponential" -> "an exponential"

**Summary Of The Paper:**

Authors propose AdaStride, a differentiable layer that learns an input-dependent and irregularly sampled downsampling.

**Summary Of The Review:**

The potential impact is large but the authors may have disregarded the main baseline they should compare against.

---

> ### Author Response · Authors · 2022-11-15
> **Response to MsFg (1/2)**
>
> We thank you for acknowledging the very large potential impact of our paper, and also for giving valuable feedback on our paper. We have incorporated your feedback into our paper and provided responses to your questions below.
> &nbsp;
> &nbsp;
> **Q: I suspect that authors do not show how close their approach is to the learnable alignment layer of Efficient-TTS, which seems to accomplish the same thing (learn an alignment between x and its downsampled version). If the models are indeed very close and authors simply apply it in the context of downsampling, the contribution is still worth publication but they should be honest about that, while currently, the reference to Efficient-TTS is kind of hidden in the text.**
> A: When we designed the vector positioning process (VP), we were inspired by Efficient-TTS so we first cited the paper, and accepting your feedback, we have updated our paper to clarify this. Thank you for helping us avoid unintentional issues. However, it was not just the application of Efficient-TTS in the context of downsampling. Unlike Efficient-TTS that aligns the text and speech modalities based on the cross-attention mechanism, VP is trained to directly calculate the relevant displacements of input vectors based only on the input vectors using the fully-connected layer and the exponential function. In addition, we proposed additional design components in the context of the downsampling, such as AdaStride-F and position loss. For example, although upsampling the text representations was not a consideration in the context of the text-to-speech, it is a consideration in the context of the downsampling so we additionally use the position loss to make the VP learning stable.
> &nbsp;
> &nbsp;
> **Q: Authors should specify more clearly how they replace strided convs by adastride, even though I assume it first applies an unstrided convolution.**
> A: Thank you for your suggestion. Yes, your thought is correct and we updated our paper to explain that part more clearly.
> &nbsp;
> &nbsp;
> **Q: AdaStride-F is presented as an implementation trick, yet performance between AdaStride and AdaStride-F varies a lot (e.g. for ASR). Why?**
> A: We conjecture this is because preserving the temporal order of the input vectors is more significant in ASR, especially for recognizing phonemes. Since there is no overlap between the pooling windows of AdaStride-F, the temporal order of the representations could be preserved better than in AdaStride.
> &nbsp;
> &nbsp;
> **Q: This approach is conceptually very different from spectral pooling or DiffStride that exploits global spatial information for downsampling, while AdaStride only repeats or removes frames, as would be done in DTW. It would be interesting to have more discussion on this fundamental difference.**
> A: As you and also the reviewer kiYh said, the operating principles of AdaStride and spectral downsampling methods are different so it would be valuable to discuss them in more detail. Therefore, we added a discussion about learning adaptive strides and anti-aliasing with additional experiments in the updated paper, so please refer to Appendix F.
> &nbsp;
> &nbsp;
> **Q: The gains on the audio classification tasks are not significant, and I find the justification for the particularly weak performance on TUT not convincing, or at least incomplete. The authors explain that the "information density is uniform" in acoustic scenes which are somehow incorrect (and speculative anyway) since a car honking or a bird chirping are transient events that can hint at the type of acoustic scene. (updated on 18 Nov. 06:05 GMT.)**
> A: Thank you for pointing out the aspect that seems not very convincing. To make the justification more convincing, we added the analysis of how effectively the AdaStride layer learns to utilize the adaptive strides according to the different datasets in Appendix E. The analysis shows that our AdaStride actually utilizes the adaptive strides to the smallest extent on the TUT urban dataset. In addition, we guess you can understand better why TUT urban dataset was less advantageous for AdaStride compared to the other datasets after listening to the audio samples, so please listen to the samples at this demo page: http://adastraudiosamples.tk/
> &nbsp;
> &nbsp;
> **Q: it is explained that AdaStride-S improves IEMOCAP but this is not what is shown in Table 1**
> A: AdaStride-S is a model that was included to see whether our AdaStride is indeed effective when its cost is almost the same as the ‘strided convolution’. Therefore, the accuracies of AdaStride-S were also compared to the accuracies of the ‘strided convolution’.
> &nbsp;
> &nbsp;

---

> > ### Author Response · Authors · 2022-11-16
> > **Response to MsFg (2/2)**
> >
> > **Q: The audio classification dataset is quite small and subject to wide variations in results based on confounding factors such as the random seed; I find it not very convincing to cross-validate configurations (from AdaStride to AdaStride-S) or hyperparameters (sigma) on them, in particular on the test set.**
> > A: Therefore, in audio classification, we tried to minimize the randomness arising from the different random seeds by training five models with different random seeds and by reporting the mean accuracies of the models. Also, when we chose the model configurations and hyperparameters, such as sigma values, we validated the model configurations on the validation split and measured the accuracies on the test split under the setting. Plus, we think that the ablation study on the effect of sigma value can also deliver much intuition to readers about choosing the appropriate sigma value.
> > &nbsp;
> > &nbsp;
> > **Q: DiffStride is excluded from ASR experiments due to learning the downsampling factor; why is it a limitation?**
> > This is because, in ASR, the downsampling methods are used for purpose of reducing the time steps by exactly 1/4. However, it is impossible to exactly control the downsampling ratio when using DiffStride because its downsampling ratio keeps varying during training. However, to answer your question, we conducted additional ASR experiments, where we combined a DiffStride downsampling layer and a SpectralPooling downsampling layer to control the total downsampling ratio to be 4, and the results are shown in the table below. In the experiments, our AdaStride still outperforms the other downsampling methods, and we will update the results.
> > &nbsp;
> > &nbsp;
> >
> > || &nbsp; &nbsp; dev-clean | &nbsp; &nbsp; &nbsp; dev-other | &nbsp; &nbsp; test-clean | &nbsp; &nbsp; test-other |
> > |-|-|-|-|-|
> > | DiffStride & Spectral Pool. | &nbsp; &nbsp; 1.48 / 5.02 | &nbsp; &nbsp; 4.78 / 12.88 | &nbsp; &nbsp; 1.60 / 4.93 | &nbsp; &nbsp; 4.62 / 12.81 |
> >
> > &nbsp;
> > &nbsp;
> > **Q: Why is Adastride limited to sequential data? What are the technical challenges to extending it to an arbitrary number of dimensions?**
> > A: Although the concept of adaptively downsampling the input representations can be extended to various domains such as images, it is not straightforward to apply AdaStride in the two-dimensional space. Especially, it is challenging to locate the spatial features onto the two-dimensional plane defined by indices based on the cumulative sum operation, while preserving their spatial ordering. To answer your question, we added the discussion about applying AdaStride to images with additional experiments in Appendix I, so please refer to it.
> > &nbsp;
> > &nbsp;
> > **Q: There are several typos**
> > A: Thank you for your suggestion and we updated our paper to reflect it.
> > &nbsp;
> > &nbsp;
> > If your concerns are sufficiently addressed and the updated paper seems to have improved by reflecting your feedback, we appreciate it if you would consider accepting our paper.  Otherwise, please let us know about your remaining concerns.

---

> ### Author Response · Authors · 2022-12-12
> **Thank you for updating the score!**
>
> Dear Reviewer MsFg,
>
> We sincerely thank you for the updated score, and we again appreciate your time and effort in reviewing our paper and reading our comments, which really helped us greatly in improving our paper.
>
> Best,
>
> Authors

---

### Official Review · Reviewer_kiYh · 2022-10-31

**Confidence:** 4
**Correctness:** 4
**Technical Novelty And Significance:** 3
**Empirical Novelty And Significance:** 3
**Recommendation:** 6

**Clarity, Quality, Novelty And Reproducibility:**

This paper is clearly written with good clarity and has made novel contributions. Source codes are provided in the supplementary materials.

**Strength And Weaknesses:**

**Strength**

1. It is interesting and novel to apply varying downsampling ratios across the same data instance to better extract the task-relevant information.

2. The proposed method is extensively validated across several audio processing tasks, indicating its general effectiveness on audio processing.


**Weakness**

1. The major concern for the proposed method is its generality for 2D inputs like images, considering all the baselines, including spectral pooling and DiffStride, also work for 2D images. Intuitively the principles discussed in Section 3.1 can be applicable for 2D images but the alignment matrix A should be defined on 2D distances and the introduced computational overhead should be quite large. A discussion for such generality is expected, otherwise the proposed method is limited to sequential data and not so general as the baselines.

2. Considering the fully connected layers in AdaStride are executed for each input token, this may incur larger overheads and limit the scalability of long sequences.

3. The author analyzes that the spectral pooling is a better choice for the TUT dataset with high-frequency noises in Table 1. Although this is reasonable, I wonder whether the proposed AdaStride can be also applied in the frequency domain on top of spectral pooling for improving the accuracy on the TUT dataset.

4. The WER improvements in Table 3 is limited. Experiments on low-resource speech, e.g., LibriSpeech-10h/100h, may lead to more extensive comparisons.

5. The margins around figures and tables are sometimes too large, e.g., Table 3 and Figure 4, which can be better organized.






**Summary Of The Paper:**

This paper proposes a new downsampling method for sequential data, which supports varying downsampling ratios across the same data instance. To achieve this, it rearranges each time step on a one-dimensional line segment, which is used to learn an alignment matrix for performing the downsampling. Extensive experiments across several audio processing tasks validate the effectiveness of the proposed method.

**Summary Of The Review:**

Considering the interestingness and the achieved experimental improvements, I tend to rank this paper as marginally above the acceptance threshold.

---

> ### Author Response · Authors · 2022-11-15
> **Response to Reviwer kiYh**
>
> We thank you for thinking our paper is interesting and novel and also for giving us valuable feedback. We have incorporated your feedback into our paper and provided responses to your questions below.
> &nbsp;
> &nbsp;
> **Q: The major concern for the proposed method is its generality for 2D inputs like images. Intuitively the principles discussed in Section 3.1 can be applied to 2D images but the alignment matrix should be defined on 2D distances and the introduced computational overhead should be quite large. A discussion for such generality is expected, otherwise, the proposed method is limited to sequential data and not so general as the baselines.**
> A: Although the concept of adaptively downsampling the input representations can be extended to various domains such as images, it is not straightforward to apply AdaStride in the two-dimensional space. Especially, it is challenging to locate the spatial features onto the two-dimensional plane defined by indices based on the cumulative sum operation, while preserving their spatial ordering. However, following your suggestion, we added the discussion about applying AdaStride to images with additional experiments in Appendix I, so please refer to it.
> &nbsp;
> &nbsp;
> **Q: Considering the fully connected layers in AdaStride are executed for each input token, this may incur larger overheads and limit the scalability of long sequences. (updated on 16 Nov. 08:35 GMT.)**
> A: Yes, there is an overhead when using AdaStride instead of strided convolution. Therefore, we also conducted audio classification experiments with AdaStride-S where we applied AdaStride downsampling layer only at the lowest resolution and saw there was much accuracy improvement with no significant computation overhead. According to this, people can decide to use our AdaStride layer only at the lowest resolution to address the scalability issue. Moreover, if it is used for the practical purpose as was in ASR experiments, the computation cost is insignificant compared to those of the main huge Transformer blocks. In addition, we added the computational complexities in Appendix G, in addition to the actual time and memory cost of the different downsampling methods. We hope this update will give intuition to future studies about reducing the overhead. Thank you for your valuable feedback and please refer to the update.
> &nbsp;
> &nbsp;
> **Q: I wonder whether the proposed AdaStride can be also applied in the frequency domain on top of spectral pooling to improve the accuracy on the TUT dataset.**
> A: I guess this question is about combining the advantages of learning adaptive strides and anti-aliasing. Yes, it can be implemented by locating a low-pass filter before AdaStride to remove the high-frequency noise. However, it then needs to choose the appropriate level of low-pass filtering because the adaptive sampling rate means different Nyquist frequencies. Actually, when we conducted the experiment of putting low-pass filters before AdaStride layers, which remove the top-half frequency components, there was not that much gain, cf. Table 8.
> &nbsp;
> &nbsp;
> **Q: The WER improvements in Table 3 are limited. Experiments on low-resource speech, e.g., LibriSpeech-10h/100h, may lead to more extensive comparisons.**
> A: Thank you for your valuable suggestion that makes the experiments more convincing. We additionally conducted the experiments that use only a small fraction of the LibriSpeech dataset, and also in the experiments, our AdaStride outperformed the other downsampling methods. This result is updated in the revision in Appendix H.
> &nbsp;
> &nbsp;
> **Q: The margins around figures and tables are sometimes too large, e.g., Table 3 and Figure 4, which can be better organized.**
> A: Thank you for your suggestion. We updated our paper to reflect it.
> &nbsp;
> &nbsp;
> If your concerns are sufficiently addressed and the updated paper seems to have improved by reflecting your feedback, we appreciate it if you would consider giving us a more positive score. Otherwise, please let us know about your remaining concerns.

---

> ### Author Response · Authors · 2022-12-13
> **Looking forward to your opinion after our rebuttal!**
>
> Dear reviewer kiYh,
>
> Thank you again for your reviews on our paper. In our response above, we have responded to your comments by providing the following results and clarifications.
>
> * We have added new analyses on applying our AdaStride on the image dataset for more discussion in Appendix I (the computational overhead of 2D alignment is involved with the next bullet).
> * We have clarified the concern about computation overheads of AdaStride by conducting an additional experiment with Adstride-S, and we have also added theoretical and empirical analyses on computations of different downsampling methods in Appendix G, where we showed that its scalability on long sequences is linear like the strided convolution.
> * To satisfy your curiosity about combining spectral pooling and AdaStride together, we have conducted an additional experiment in Appendix F.
> * Taking your advice, we have conducted more ASR experiments on low-resource environments in Appendix H, where our AdaStride consistently outperformed the other downsampling methods.
> * We reorganized the figures and tables more neatly.
>
> We are wondering whether these results and clarifications have addressed your questions and concerns. Please express your opinion through comments or by updating your score before the deadline.
>
> Thank you!
>
> Authors

---

> > ### Comment · Reviewer_kiYh · 2022-12-13
> > **Reply to Author Response**
> >
> > Thank you for the additional experiments and further clarifications. Although most of my concerns are properly resolved, the limited generality to 2D cases somewhat constrains the scope and value of the proposed method. In addition, more ablation studies based on ASR tasks in addition to the newly added ones in the appendix could strengthen the evaluation of this paper. I will keep my original score for now.

---

### Author Response · Authors · 2022-11-17
**General comments to the reviewers**

We thank all reviewers for the positive evaluations on the novelty of our method, clarity of writing, the potential impact, and the extensive experiments.

Also, your feedback was really helpful in making our paper more convincing and inspiring, as well as the positive evaluations.

During this rebuttal period, we have updated the paper further by incorporating the suggestions.
&nbsp;
&nbsp;
**Below is a summary of the changes:**
* We added a **'Future work' paragraph** before the Conclusion section, with **two additional Appendices G, I**. They are added to discuss the limitation of our method such as **generality in 2D images** and **computation cost** (reviewer kiYh, PVam, PyHD).
* We added a **discussion on the relationship between learning adaptive strides and anti-aliasing (Appendix F)**. It gives information about **how to combine their advantages** and **how shift-invariance our AdaStride is** (reviewer kiYh, MsFg, PVam).
* We added an experiment of **speech recognition conducted in a low-resource setting (Appendix H)** to make the ASR experiments more convincing (reviewer kiYh).
* We added **Appendix E** to give an analysis of how effectively our AdaStride utilizes the adaptive strides according to datasets in audio classification tasks (reviewer MsFg, PVam).
* We incorporated **other suggestions for improving our paper** (e.g., improve the clarity of the paper, fix several typos, and write the captions of the tables and figures in more detail) (reviwer kiYh, MsFg, PVam).

---

### Author Response · Authors · 2022-11-18
**Gentle reminder for discussion.**

**Dear reviewers,**

Thank you again for your valuable and constructive feedback. They were insightful and allowed us to improve our paper through additional experiments and clarifications. Approaching the deadline, we'd like to remind the reviewers of the fact that the updated version of the paper has significant improvements with respect to most of the comments pointed out by the reviewers. So, please check out our general comments to all reviewers and the individual responses where we address specific questions (there can be an update after the first post, so please check it together). Please also let us know if there are any additional clarifications or experiments we can provide to show the merit of our paper.

Thank you, and looking forward to hearing from all of you!

---

### Author Response · Authors · 2022-12-07
**Now, the deadline for this discussion is within a week.**

**Dear reviewers,**

We write this comment to remind you that now there is less than a week left for this discussion period.

We sincerely hope to be able to receive your response to our rebuttal for this discussion period to be more valuable.

 If your concerns are sufficiently addressed and the updated paper seems to have improved by reflecting your feedback, we appreciate it if you would consider giving us a more positive score.

Otherwise, please let us know about your remaining concerns.

We again thank you for your valuable feedback :D

---

> ### Author Response · Authors · 2022-12-12
> **There's only one day left until the deadline, and we'd really appreciate it if you could express your opinion after our rebuttal.**
>
> This is a friendly reminder that **there's only one day left until the deadline**, and we'd really appreciate it if you could express your opinion after our rebuttal.
> &nbsp;
> &nbsp;
> Especially, **the metareview due is also at the end of this discussion stage**, so please express your opinion a little earlier if possible so that all the revisions of the paper and all reviewers' opinions are adequately reflected in the final decision within a given period of time.
> &nbsp;
> &nbsp;
> Thanks a lot again, and with sincerest best wishes
> Authors

---

### Author Response · Authors · 2022-12-13
**Summarization of Rebuttal**

We thank the reviewers and chairs for their efforts, and are very pleased for the reviewers to acknowledge the strengths of our work:
**'the novelty of our method', 'clarity of writing', 'the large potential effectiveness and impact as a plug-in downsampling module', and 'the extensive experiments'.**

According to their comments, there were several concerns that are commonly shared by the reviewers and we have addressed the concerns as follows:

**How to extend AdaStride to datasets having higher dimensions such as images?**
To address this concern, we added Appendix I in the revision, where we conducted image classification tasks using our AdaStride. In the experiments, we showed that AdaStride can still learn effective adaptive strides by showing that it outperformed the strided convolution and spectral pooling, and the attention weights actually had higher values near the target object region. Furthermore, even though it underperformed DiffStride (the previous state-of-the-art pooling method) in the case of images, we tried to give a direction for future work for improving AdaStride by explaining why extending our AdaStride to the higher dimensions is not straightforward and which problem remained.

 **There should be a comparison of the computation costs between different downsampling methods**
To address this concern, we added Appendix G in the revision, where we first theoretically analyzed the computational complexities of different downsampling methods. According to this, although our AdaStride has quadratic computation complexity, we also showed that the AdaStride-F has linear computation complexity. It indicates that our method is superior to the spectral-based pooling methods and has the same computation complexity as the strided convolution. Then, we also compared the computation costs of the different downsampling methods by practically measuring the computational costs that were actually used in the audio classification task.

 **Several recommendations on additional experiments**
To address these concerns, we have added a lot of additional experiments in the revision: (1) we evaluated the accuracy changes while shifting the inputs to see the robustness of AdaStride to the input shift; (2) we additionally conducted the ASR experiments in low-resource setting to make the experiments more convincing; (3) we conducted an experiment of combining spectral pooling and AdaStride to have a discussion on the fundamental difference between spectral based pooling and learning adaptive strides; (4) we analyzed the effectiveness of AdaStride according to different datasets to see in which condition our AdaStride is especially effective; (5) we added the performance of DiffStride in ASR by combining it with a Spectral pooling layer for a fair comparison.

In addition to these, we could also clarify many concerns thanks to the detailed comments of the reviewers. Even though this rebuttal process will end very soon, and it would have been better if we could address the concerns that might remain, we hope to have an opportunity to discuss them later.

We again thank all the reviewers and chairs for taking your precious time for this rebuttal process.

Sincerely,
Authors of Paper 1013

---

### Decision · Program_Chairs · 2023-01-20

**Decision:**

Reject

**Justification For Why Not Higher Score:**

Reviewers raised some concerns about quadratic time complexity of AdaStride; although authors propose a variant AdaStride-F which has linear complexity but reviewers noted in the virtual meeting that performance is not so good. A major concern that came up in the virtual meeting with reviewers was that the application scope is limited to 1D data with major focus on audio classification tasks in the paper and ASR experiments on Librispeech does not use full data. Reviewers believed that these points make the paper fall slightly below the acceptance bar for ICLR.

**Justification For Why Not Lower Score:**

N/A

**Metareview: Summary, Strengths And Weaknesses:**

The paper proposes a new adaptive downsampling method for sequential data. Adaptive strides allow for preserving more information for task-relevant parts by using smaller strides for those and larger strides for less important parts, while preserving a global downsampling factor. Experiments across several audio processing tasks validate the effectiveness of the proposed method on audio data. Reviewers raised some concerns about quadratic time complexity of AdaStride; although authors propose a variant AdaStride-F which has linear complexity but reviewers noted in the virtual meeting that performance is not so good. A major concern that came up in the virtual meeting with reviewers was that the application scope is limited to 1D data with major focus on audio classification tasks in the paper and ASR experiments on Librispeech does not use full data. Reviewers believed that these points make the paper fall slightly below the acceptance bar for ICLR.